# Still waters run deep in large-scale genome rearrangements of morphologically conservative Polyplacophora

Julia D Sigwart[1,2]*[†], Yunlong Li[3,4]*[†], Zeyuan Chen[1,2], Katarzyna Vončina[1,2], Jin Sun[3,4]*

[1]Department of Marine Zoology, Senckenberg Research Institute and Natural History Museum Frankfurt, Frankfurt, Germany; [2]Institute of Ecology, Evolution & Diversity, Goethe University, Frankfurt, Germany; [3]Key Laboratory of Evolution & Marine Biodiversity (Ministry of Education) and Institute of Evolution & Marine Biodiversity, Ocean University of China, Qingdao, China; [4]Laboratory for Marine Biology and Biotechnology, Qingdao Marine Science and Technology Center, Laoshan Laboratory, Qingdao, China

**\*For correspondence:**
julia.sigwart@senckenberg.de (JDS);
liyunlong@ouc.edu.cn (YL);
jin_sun@ouc.edu.cn (JS)

[†]These authors contributed equally to this work

**Competing interest:** The authors declare that no competing interests exist.

## eLife Assessment

This **important** study advances our understanding of genome annotations for chiton genomes. It provides a **solid** estimation of syntenic relationships for the chromosomes of the four new genomes plus an analysis linking these to other available chiton genomes, and an update for how these relate to molluscan genomes.

**Abstract** A major question in animal evolution is how genotypic and phenotypic changes are related, and another is when and whether ancient gene order is conserved in living clades. Chitons, the molluscan class Polyplacophora, retain a body plan and general morphology apparently little changed since the Palaeozoic. We present a comparative analysis of five reference quality genomes, including four de novo assemblies, covering all major chiton clades, and an updated phylogeny for the phylum. We constructed 20 ancient molluscan linkage groups (MLGs) and show that these are relatively conserved in bivalve karyotypes, but in chitons they are subject to re-ordering, rearrangement, fusion, or partial duplication and vary even between congeneric species. The largest number of novel fusions is in the most plesiomorphic clade Lepidopleurida, and the chitonid *Liolophura japonica* has a partial genome duplication, extending the occurrence of large-scale gene duplication within Mollusca. The extreme and dynamic genome rearrangements in this class stands in contrast to most other animals, demonstrating that chitons have overcome evolutionary constraints acting on other animal groups. The apparently conservative phenome of chitons belies rapid and extensive changes in genome.

## Introduction

The genomic mechanisms that enable or limit the evolution of major morphological changes remains one of the great questions of evolutionary biology. Living members of Mollusca represent the broadest morphological disparity of any animal phylum: mollusc body plans encompass squid,

worms, living rocks and candy-coloured tree snails, as well as diverse intermediate and additional novel forms known from the fossil record. Early work on chromosome numbers in meiotic division used molluscs, and interest in understanding patterns in chromosome numbers across species also led to fundamental insights in animal polyploidy. Despite playing a key role in early advances, this important phylum has lagged in terms of quality and taxonomic coverage of whole genome sequence data (*Chen et al., 2025*). Reconstructing the evolution of genome architecture through deep time in diverse molluscs remains critical to understand genome evolution in animals. High-quality genomic data for the deeply divergent, morphologically constrained chitons, would be expected to offer an opportunity to explore ancient genetic traits and evolutionary mechanisms preserved across the long span of animal evolution.

Prior work has reasonably assumed that rates of intra-chromosomal gene translocation are constant within major groups (*Mackintosh et al., 2023*). If this is true, syntenic rearrangement could be a clocklike indicator of divergence times. But rates of genomic rearrangement are not well known in molluscs, nor how they might vary across this vast clade, and higher rates of rearrangement confound reconstruction of ancestral states. In molluscs, an excellent fossil record allows an independent record of divergence times that will provide more insight into the variability (and utility) of rates of rearrangement as a measure of divergence time.

Chitons have long been considered the key taxon to understanding the ancestral molluscan body plan (*Wanninger and Wollesen, 2019*). Over 1000 living chiton species worldwide all possess an eight-part shell armour that has remained superficially unchanged with relatively little variation since the Carboniferous, over 300 Million years ago (*Sigwart, 2017*). Chitons are increasingly important for bio-inspired design, which will benefit from genomic tools to understand genetic control of their flexible armour and unique sensory systems (*Ampuero et al., 2024*; *Varney et al., 2021*). Chitons are generally conserved, yet the known species richness in chitons is higher than for the more morphologically and behaviourally diverse extant cephalopods. The adaptations that drive speciation in this group remain an open question.

One key issue is whether polyplacophoran molluscs are conservative per se or whether their diagnostic body plan adaptations are so distinctive and strange that this has overshadowed full understanding of additional adaptive traits: fully innervated shells, iron mineralised radula, living at almost all bathymetric depths and latitudes of the world ocean. While the body plan of chitons has persisted for over 300 My, this is a template for remarkable adaptations that have only recently begun to be appreciated.

Chitons are grazers and most are not apparently highly ecologically specialised (*Sigwart and Schwabe, 2017*). These animals mostly have separate sexes and all are broadcast spawners, are not migratory, cannot reasonably be subject to strong sexual selection, and many similar species co-occur in parapatric or sympatric radiations (*Kelly and Eernisse, 2008*). Previously published karyotype data shows closely related species with overlapping ranges can have different numbers of chromosomes, such as *Acanthochitona discrepans* (1n=8) and *A. crinita* (1n=9) in the North Atlantic (*Certain, 1951*). Chromosome rearrangement presents an attractive speculative explanation that could support maintaining species boundaries in this clade.

Here, we sequenced four new reference-quality genomes that cover the three taxonomic orders of living chitons: *Deshayesiella sirenkoi* (Lepidopleurida), *Callochiton septemvalvis* (Callochitonida), *Acanthochitona discrepans* and *A. rubrolineata* (Chitonida). Callochitonida is sister to Chitonida (*Moles et al., 2021*) and together these could be considered the Chitonida sensu lato. One aim here was to test the potential genomic differences separating these two orders within Chitonida s.l. We used these four new genomes together with previously published genomic data to reconstruct a phylogeny for the phylum. This allows us to confidently reconstruct ancestral chromosome arrangement of total-group Mollusca, and at different transitions within Polyplacophora.

## Results

We sequenced chromosome-level genomes of *Deshayesiella sirenkoi* (Lepidopleurida) from the Daikoku vent field, Western Pacific Ocean, *Callochiton septemvalvis* (Chitonida s.l.: Callochitonida) and *Acanthochitona discrepans* (Chitonida sensu stricto) both from the intertidal of Strangford Lough, N. Ireland, and the congener *A. rubrolineata* from the intertidal of Qingdao, China. These were sequenced with PacBio HiFi and scaffolded using Hi-C, resulting in high-quality assemblies with over

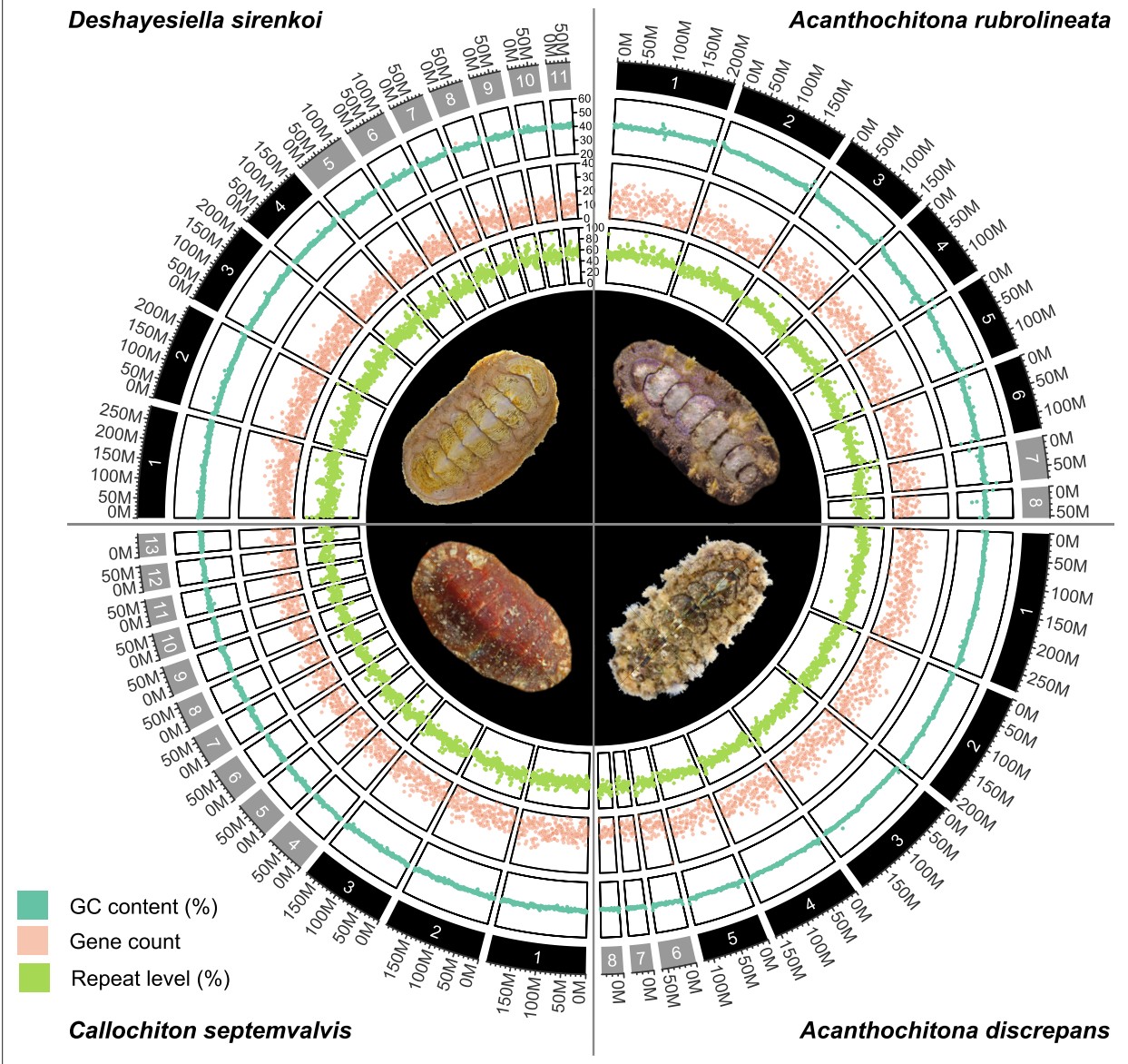

**Figure 1.** CIRCOS plots for four new chitons genome assemblies, clockwise from top left: one species in the order Lepidopleurida *Deshayesiella sirenkoi*, and three species in the clade Chitonida sensu lato*: Acanthochiton rubrolineata, A.discrepans*, and *Callochiton septemvalvis*. Each quarter circle shows the pseudochromosome content for each species, in order of size, with concentric rings indicating GC content, gene count, percent repeat content, and a photograph of the respective species.

97% BUSCO completeness (94% for *Callochiton septemvalvis*) and with numbers of chromosomes that differed among genera from 8 to 13 (***Figure 1***, ***Appendix 1—figure 1***). These species show high levels of heterozygosity, ranging from almost 1% in *Deshayesiella* to 4.12% in *Callochiton* (***Figure 1***, ***Appendix 1—figure 2***).

Our phylogenetic results confirm the placement of chitons as sister to a monophyletic Conchifera, and we recover the expected topology within Polyplacophora consistent with other recent work using genomic and morphological characters (***Figure 2***). Within Conchifera, we confirm the topology of other recent studies ***Song et al., 2023***; however, our supplementary analyses recovered Scaphopoda sister to Gastropoda but with lower support (***Appendix 1—figure 3***). Comparison with genomes from other molluscan classes shows the molluscan ancestor had a genome composed of 20 linkage groups (***Figure 2***), we refer to these as the Molluscan Linkage Groups (MLG) 1–20. Three important fusion events are apparent synapomorphies for Polyplacophora, present in all living chitons and no

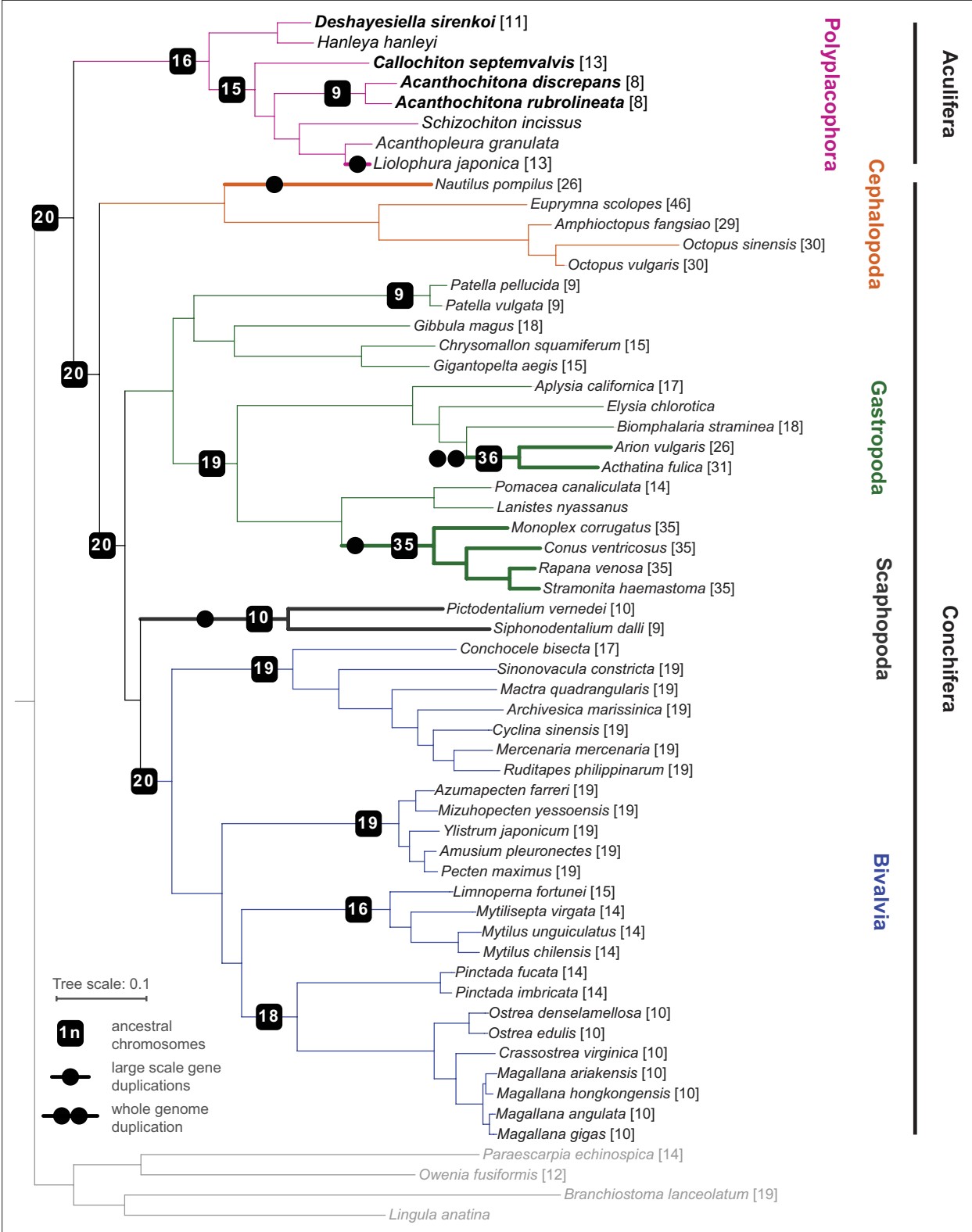

**Figure 2.** Phylogeny of Mollusca, with new genomes noted in bold type, and the chromosome number in square brackets for each species where known. Lineages with known whole (double circle) or partial genome (single circle) duplication are in thicker lines for emphasis; boxes on branches show the reconstructed ancestral 1 n chromosome number for the respective clade.

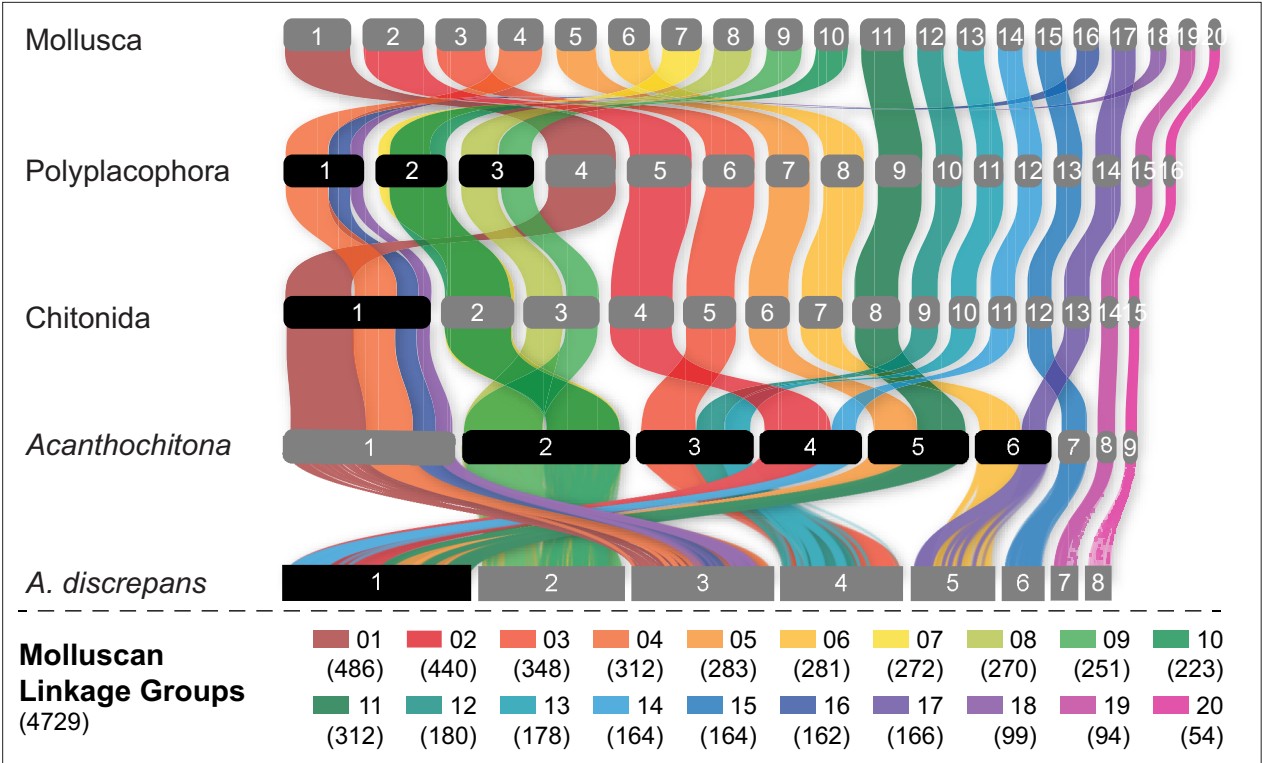

**Figure 3.** Evolution of the ancestral molluscan linkage groups (MLGs) within Polyplacophora using the lineage leading to *Acanthochitona discrepans* as an example. MLGs are distinguished by colours, at the top of the diagram and in the key at bottom showing the number of orthologs. Each row is the reconstructed karyotype of the ancestor of living Polyplacophora, the order Chitonida sensu lato, and the genus *Acanthochitona*. Reconstructed chromosomes on each row are numbered in order of size from largest (left) to smallest (right); chromosome fusions are highlighted with chromosome numbers in black boxes. This presentation highlights the extent of shifts especially in comparison to the molluscan or polyplacophoran ancestor.

Conchifera: MLG 4+16 + 18, MLG 7+10, and MLG 8+9 (*Figures 3 and 4*, *Appendix 1—figures 4–7*). There are additional fusions and intra-chromosomal rearrangements that are different in every species sampled.

## Discussion

### Chiton genomes show frequent and extreme rearrangement

Four new chiton genomes represent the most complete genomes sequenced for the class, adding to several previously published partial or complete genomes (*Hui et al., 2024*; *Liu et al., 2023*; *Varney et al., 2022*; *Varney et al., 2021*; *Chen et al., 2025*; *Supplementary file 1, table S3*). Our comparisons of conchiferan and aculiferan (polyplacophoran) linkage groups confirm previous studies that also predicted a haploid karyotype of 20 for the molluscan ancestor based on other metazoans (*Simakov et al., 2022*).

Previous work demonstrated the variability in chiton karyotypes; species in the clade Chitonida have reported haploid numbers ranging maximally from 6 to 16 (*Odierna et al., 2008*) with a mode of 11 (*Hallinan and Lindberg, 2011*). Our data give the first indication for expected chromosome count in a species in Lepidopleurida (1n=11) within this established range. Using five chromosome-level genome assemblies for chitons, we reconstructed the ancestral karyotype for Polyplacophora (more strictly the taxonomic order Neoloricata), and all intermediate phylogenetic nodes to demonstrate the stepwise fusion and rearrangement of gene linkage groups during chiton evolution (*Figure 3*).

Chitons demonstrate extreme genome rearrangement, even within a single genus. This represents not only gene order differences but syntenic (co-occurrence) changes in genomic architecture. Species of the genus *Acanthochitona* have a relatively short divergence time of maximally ~23 My based on the fossil record (*Dell'Angelo et al., 2020*). *Acanthochitona discrepans* and *A. rubrolineata* each have eight haploid chromosomes but these result from two different fusions compared to the reconstructed

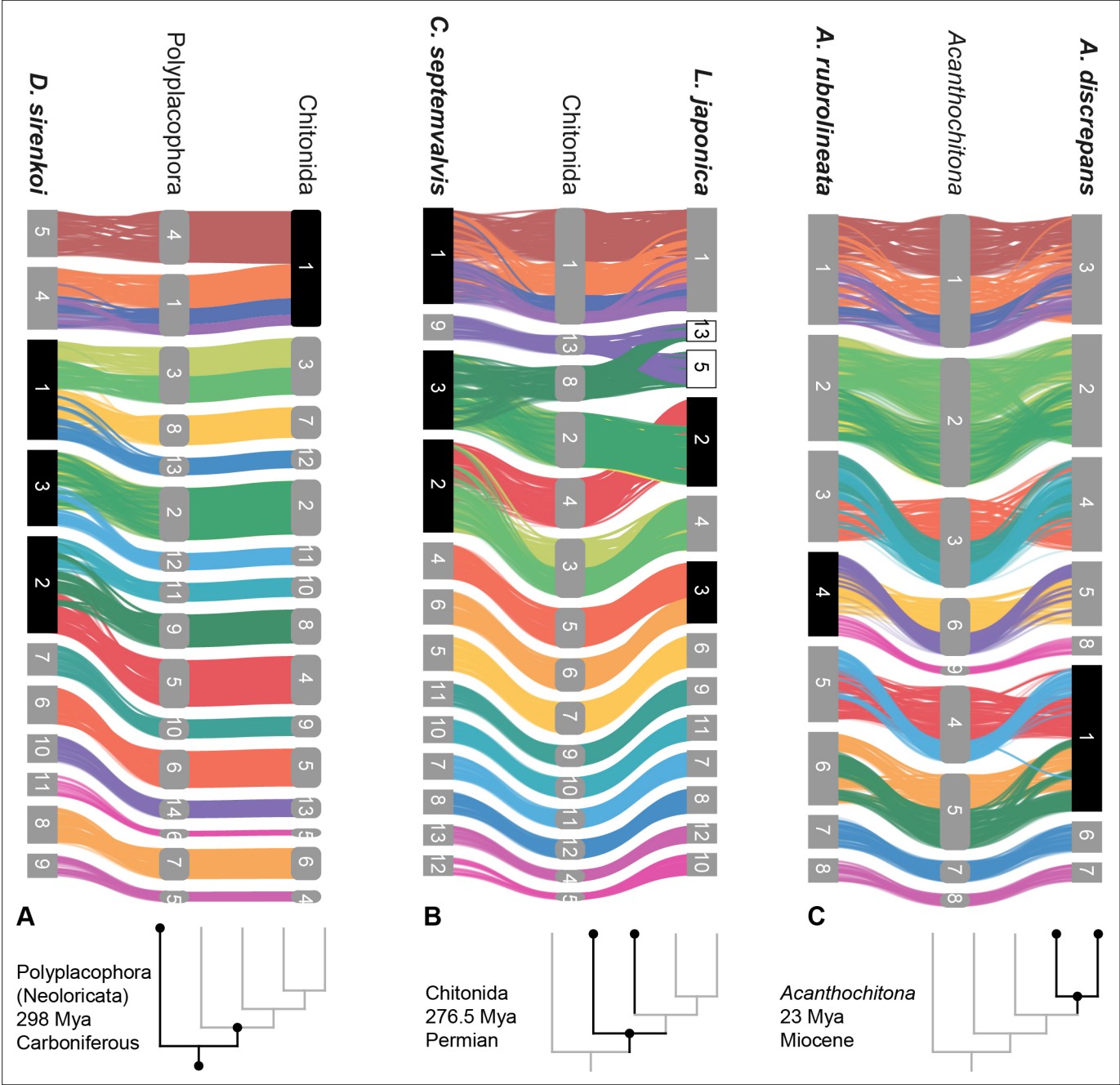

**Figure 4.** Syntenic rearrangements of MLGs within the evolution of Polyplacophora. Each part shows the reconstructed karyotypes of an ancestor (middle) and two descendent lineages, with a schematic cladogram for orientation. From left to right, the divergence of **A** ancestor of Polyplacophora, leading to the lepidopleuran species *Deshayesiella sirenkoi* (left) and ancestor of Chitonida (right), **B** ancestor of Chitonida s.l. leading to the callochitonid *Callochitona septemvalvis* (left) and chitonid *Liolophura japonica* (right) and **C** ancestor of the genus *Acanthochitona* leading to the two congeneric species *A. rubrolineata* (left) and *A. discrepans* (right). Colours and presentation are as in *Figure 3*, and chromosome numbers indicate the sequence in terms of size from largest (1) to smallest. Here, the chromosomes are not in order of size but reordered such that each transition from the nearest ancestor is visible in more detail. Chromosome fusions are highlighted with chromosome numbers in black boxes, and duplications in white boxes.

ancestral karyotype of *Acanthochitona* (*Figures 1 and 4C*). Previous karyotype data for *A. discrepans* were actually based on specimens of *A. crinita*; the species are very similar but the correct identification can be determined based on the geographic distributions (*Vončina et al., 2023*). *Acanthochitona crinita* has nine haploid chromosomes (*Certain, 1951*): this is yet another syntenic arrangement within

the same genus. There are major changes between congeners in different ocean basins (the Pacific *A. rubrolineata*) but also between two species in the NE Atlantic (*A. discrepans* and *A. crinita*) that are morphologically and ecologically almost indistinguishable.

The largest number of novel fusions among chiton genomes is three, in the lepidopleuran *Deshayesiella* (*Figure 4A*). This in particular contrasts with the naive expectation that gene arrangement in most chiton genomes would be relatively conserved. Living species of Lepidopleurida retain more plesiomorphic morphology and this clade has a deep fossil record extending to the lower Carboniferous (*Ampuero et al., 2024*; *Sigwart, 2017*); yet the exemplar of this order shows the most deviations from the reconstructed ancestral karyotype.

The remaining living chitons (Chitonida s.l.) comprise two sister clades recognised as separate orders: Chitonida and Callochitonida (*Moles et al., 2021*). *Callochiton* also has two additional fusion events, and the chitonid *Liolophura japonica* has a partial genome duplication, with two linkage groups fused apparently at first and then duplicated (*Figure 4B*). In our reconstruction of ancestral karyotypes, there is no differences in arrangement between the ancestor of Chitonida sensu stricto or the ancestor of Chitonida sensu lato (Chitonida + Callochitonida). The four species in Chitonida s.l. share a large, fused chromosome (MLG 01+04 + 16+18) that is notably not well-mixed in *Liolophura japonica* (i.e. Chr01 in *Liolophura japonica*, Chr01 in *Callochiton septemvalvis*, Chr01 in *Acanthochitona rubrolineata*, and Chr03 in *A. discrepans*). Part of this pattern has its origin in the ancestral chiton karyotype and is retained in the lepidopleuran *Deshayesiella* (i.e. MLG 04+16 + 18). This implies a variable rate of intra-chromosomal rearrangement, with several of the MLGs conserved.

## Ancestral karyotypes for Mollusca

Chromosome numbers are not strongly conserved in animals. Changes in chromosome numbers are hypothesized to be an important driver of the diversification of Lepidoptera, with a strong correlation shown between rates of chromosome number changes and speciation (*de Vos et al., 2020*). The instability in lepidopteran chromosome numbers has mainly been studied from karyotype data but recently confirmed in genomic data (*Chen et al., 2019*), and changes within genera are known to encompass both neutral and adaptive evolution (*Lucek, 2018*; *Vershinina and Lukhtanov, 2017*). However, the level of rearrangement may be less than that in Polyplacophora.

Even within a single species, chromosome fusions are not uncommon, with Roberstonian translocations occurring in roughly 1 out of every 1000 live human births (*Wilch and Morton, 2018*). A previous study speculated that chromosome loss in related clades of chitons may be the result of Robertsonian translocation (*Odierna et al., 2008*). Although we still lack information on the telomere, this simple mechanism is not a sufficient explanation for several more complex intrachromosomal rearrangements (e.g. MLG 7+10: *Figure 4*). Chromosome fusion does not present any immediate reproductive barrier, yet chromosome rearrangements between sister species can act as Dobzhansky-Muller incompatibilities and generate reproductive isolation (*de Vos et al., 2020*). Chitons are broadcast spawners, so such barriers may be speculatively advantageous, but so too are bivalves where these syntenic rearrangements are not found.

In order to compare syntenic changes within Polyplacophora to other mollusc clades, we re-analysed other available molluscan genomes in context of the newly identified MLGs. For example, the divergence of the bivalve clade Imparadentia (Lucinida +Venerida) is estimated in the Silurian, ca. 430 Mya (*Crouch et al., 2021*), or more than 150 My deeper than the divergence of crown group Polyplacophora, yet species within Imparadentia have almost no syntenic rearrangement (*Appendix 1—figures 7 and 8*). Bivalves in Pteriomorphia, which have an even deeper origin in the lower Ordovician, ca. 485 Mya, show some rearrangement but members of two orders (Arcida and Pectinida) are highly similar, while two species in Ostreida that are intensively cultivated differ from these others and from each other (*Appendix 1—figure 8*).

Potential adaptive roles may be connected to contrasting mobility of different MLGs. The smallest two MLGs are the most conserved across the five chiton genomes; MLG20 is the only group that remains separate in all bivalve and chiton taxa. This region is also not duplicated in partial genome duplications in Neogastropoda, but is duplicated in whole genome duplication in two hermaphroditic terrestrial gastropods (*Achatina fulica* and *Arion vulgaris*; *Figure 5*, *Appendix 1—figure 5*).

Earlier models based on karyotype data predicted three whole genome duplication events in the evolutionary history of living molluscs: in Neogastropoda, Stylommatophora, and coleoid cephalopods.

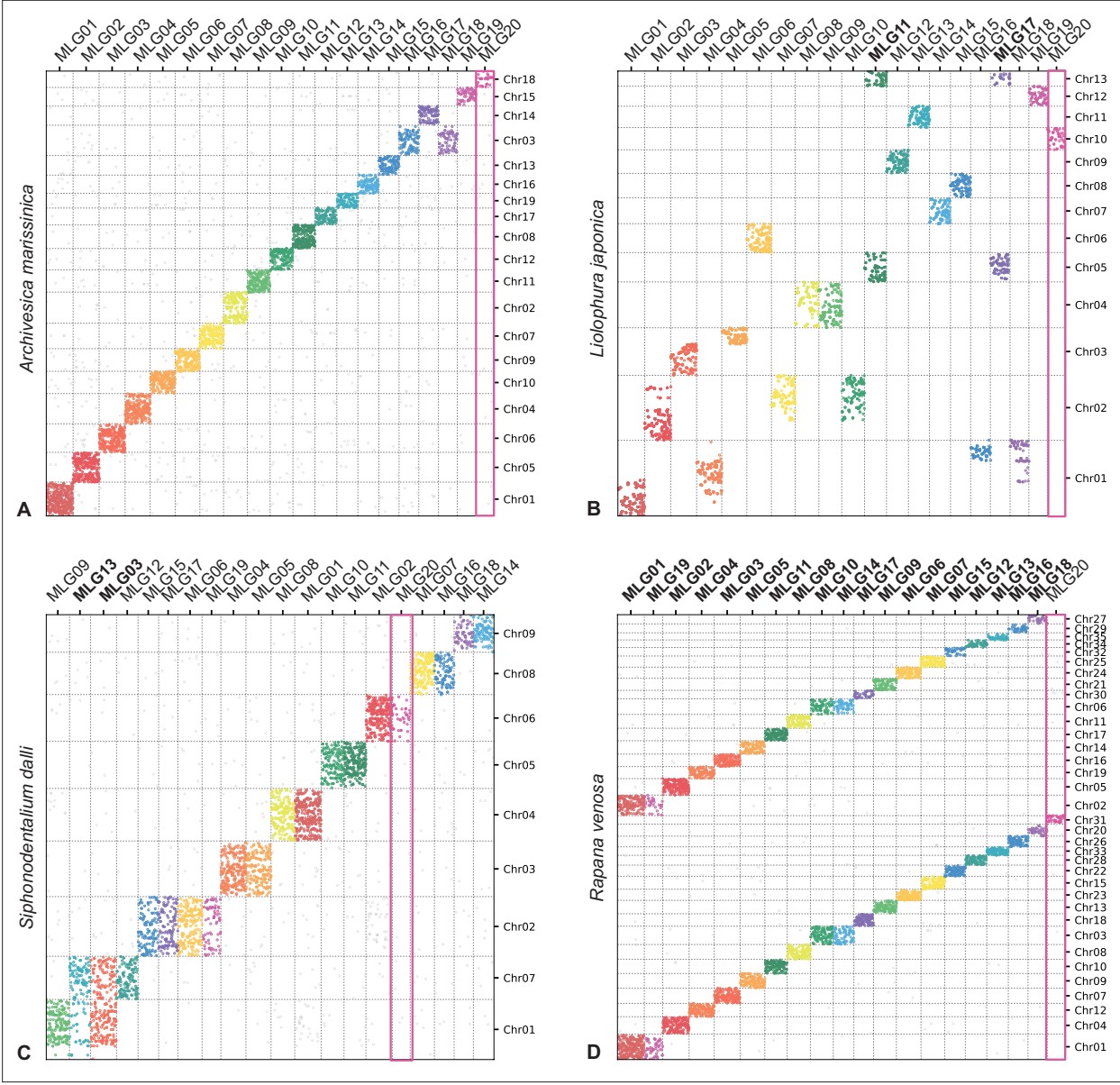

**Figure 5.** Oxford plots comparing gene occupancy of species from four different classes of molluscs (vertical) to the ancestral molluscan linkage groups (MLG, horizontal): (**A**) the bivalve *Archivesica marisinica* retains a plesiomorphic karyotype reflecting the 20 MLGs, (**B**) the chiton *Liolophura japonica* has large scale gene duplication in MLG11 and MLG17 on separate pseudochromosomes, (**C**) the scaphopod *Siphonodetalium dalli* has large scale gene duplication of MLG13 and MLG03 on separate pseudochromosomes, (**D**) the gastropod *Rapana venosa* demonstrates a nearly whole genome duplication with MLG20 not duplicated. Colours follow the presentation in *Figure 3*.

Available genomes of Stylommatophora confirm whole genome duplication, as well as large scale gene duplications in Neogastropoda (*Appendix 1—figure 6*). In coleoid cephalopod genomes, the co-occurrence of loci from MLGs and intensive fusions (*Appendix 1—figure 6*) might not be easily resolved but may be more likely from the chromosome-disrupted processes reported in previous work (*Albertin et al., 2022*). By contrast, in *Nautilus* the linkage group that contains the conserved *hox* gene sets (Chr11, MLG16) shows no signal of co-occurrence with other MLGs or on other chromosomes (*Appendix 1—figure 6*). New results also show a partial and ancient genome duplication in the chiton *Liolophura* (*Figures 2 and 4*). Given that the MLGs are well conserved in all chitons if not their order (*Appendix 1—figure 4*), the event in *Liolophura japonica* is most likely a duplication instead of fission. Whole or partial genome duplication is now known from four molluscan classes, with contrasting patterns of ploidy (e.g. gastropods) or tandem (e.g. scaphopods) duplications (*Figure 5*).

Reconstructing genome evolution is naturally more difficult for groups with high rates of intra-chromosomal rearrangement (*Farré et al., 2019*; *Muffato et al., 2023*). These patterns of co-occurrence of loci on the same chromosomes (synteny) should also persist for longer in evolutionary time, compared to faster rates of change in gene order (*Damas et al., 2022*; *Simakov et al., 2022*). Syntenic changes do not necessarily follow overall rates of translocation, which do not differ obviously among molluscs (*Supplementary file 1, table S5*). Confounding effects of rapid gene order or linkage group changes are not limited to issues of phylogenetic distance, when even closely related species have significant syntenic differences. Nonetheless, a focus on linkage group arrangements is a promising direction for the many unresolved questions of deep molluscan phylogeny. Recent studies have championed the importance of synteny-based studies on genome evolution, as a basis to understand deep divergences and also the general mechanisms that underlie genomic architecture, fusion, fission, and translocation (*Mackintosh et al., 2023*). A general trend in insect and vertebrates is for increasing chromosome numbers in derived lineages. Fusion events may be more common in more recently derived chitons or connected to specific adaptations.

## Conclusions

The relatively small number of reference quality genomes for most molluscan groups is a temporary limitation to these analyses. High heterozygosity seems to be very common in molluscs and is clearly a feature of chitons, which causes difficulty for high quality genome assembly. The heterozygosity for *Callochiton* at 4.12% far exceeds the 2.95% genomic heterozygosity reported as 'one of the highest' for Lepidoptera (*García-Berro et al., 2023*). Large-scale analyses based on genomics are equivocal about the drivers of genetic diversity, and further data from diverse clades must be included.

The concept of 'diverse groups' is mostly based on perceived variability, typically manifested via morphological, ecological, or genetic differences. As one interesting comparison, Lepidoptera is commonly regarded as super-diverse, but also represent a recognisably constrained form, with variation in striking difference in colours, patterns and wing shapes. While lepidopteran variability is visually conspicuous, chitons are apparently equally variable, as demonstrated through their genetic rearrangement. Chitons, despite exceeding 1000 species in diverse ecological niches, with a wide range of morphological adaptations, are considered a 'minor' and 'neglected' clade. The group is taxonomically challenging – on the one hand, because of the supposed morphological stasis as a group, on the other, because of their high interspecific variability. Our comparative analyses suggest chromosomal-level changes are a pattern throughout much of chiton evolution, since chromosomal rearrangements are found when comparing congeneric species (*Acanthochitona* spp., *Figure 4C*) and also across orders (*Figure 4A and B*).

Chitons exhibit chromosome rearrangements at an almost unprecedented level. This is clear in the present study, the first comparative genomic study for Polyplacophora, compared to orders of magnitude more data for better studied groups. All this poses a more general question on how we define variability, how we perceive it, and also, if we actually understand it at all? Recognition of variability in the genome and phenome is crucial in understanding and re-evaluating unbiased measures of diversity in overlooked groups of organisms. The relationship revealed between the unstable arrangement of chiton genomes and species diversity provides new insights into potential mechanisms for speciation and broader diversification within Mollusca.

## Methods

Specimens were collected alive and flash frozen in liquid nitrogen (*A. rubrolineata*) or frozen at –80 °C and stored at –80 °C. High-molecular-weight genomic DNA was extracted from the foot of an individual specimen of each species, following the guidelines of the SDS method, and used for PacBio high fidelity (HiFi) sequencing and Hi-C library preparation. The PacBio Sequel II / IIe instrument was used for sequencing in CCS mode. For transcriptome sequencing, five separate tissues were dissected from the same specimen as DNA extraction, and preserved in RNAlater: foot (F), perinotum (P), radula sac (R), shell edge (S), and visceral mass (V).

Adapters in the reads were checked and removed using HiFiAdapterFilt version 2.0. The genome survey was finished using jellyfish v2.2.10 and Genomescope v2.0 (*Ranallo-Benavidez et al., 2020*), implemented 21- and 23-mer, which produced the estimated genome size and heterozygosity level

for each species (*Appendix 1—figure 2*). The four genomes were assembled de novo based on qualified HiFi reads using hifiasm v0.16.1 (*Cheng et al., 2021*), with some additional curation steps for *C. septemvalvis* because of the high heterozygosity (Appendix 1). Potential contamination in the genome was detected and removed using blobtools v1.1.1 (*Laetsch and Blaxter, 2017*). Duplicated haplotigs and overlaps were removed for *D. sirenkoi*, *A. discrepans*, and *A. rubrolineata* using Purge_Dups v1.2.6 (*Guan et al., 2020*).

The aligner STAR version 2.7.10 a or STAR v2.7.3a was employed to map RNA-seq data into the working genome data (*Dobin et al., 2013*). The resulting alignment file served as an important support in prediction methods. Ab initio gene prediction was performed on the repeat-masked assembly in comparison with other published genomes. Summary information on assembly, gene model, and annotation is provided in *Supplementary file 1, table S2*.

We identified putative orthologous sequences shared among genomes or transcriptomes for 58 molluscs and 4 additional metazoans to generate an alignment for phylogenetic analysis using VEHoP (*Li et al., 2024*). The alignments were removed if the overlap among them was less than 20 amino acids and there were less than 75% of taxa sampled, then, each alignment was used to construct 'approximately maximum likelihood' tree using FastTree version 2.1.11 (*Price et al., 2010*). Phylogenetic relationships were investigated using IQ-TREE version 2.1.3 (*Minh et al., 2020*) with the '-MFP' model to compute the best-fit model of each partition and 1000 ultrabootstraps to test the topological support. To test the impact on conchiferan topology, we ran a second analysis excluding Solemyida, the earliest diverging clade in Bivalvia (*Appendix 1—figure 3*).

The method for identifying ancient linkage groups followed previously published works. We reconstructed the linkages of the orthologues in molluscs, with the demonstrated commands, and conserved and representative proteins and their presumptive locations (https://github.com/ylify/MLGs; copy archived at *Li, 2025*). In detail, the ancient and conserved linkage groups were inferred from seven chromosome-level mollusk assembles, including one gastropod (*Gibbula magus*), one bivalve (*Mizuhopecten yessonensis*), and the five chitons (*A. rubrolineata*, *A. discrepans*, *C. septemvalvis*, *D. sirenkoi*, and *Liolophura japonica*). The obtained gene set consisted of 4729 homologs, which could be detected in all 7 assembles and located in 20 linkage groups. The ancestral states of nodes were predicted to trace the karyotype evolutionary route in Polyplacophora, from the common ancestor of Mollusca to the genus *Acanthochitona*. Similarly, the nodes with more than two chromosome-level genomes within Mollusca were also investigated for the chromosome number of their common ancestor. We also compared the translocation rates across Mollusca, based on the non-syntenic rate of change divided by estimated divergence time. In this study, 25 genomes from four classes were selected to calculated the translocation rate at the inter-chromosome level, including 2 scaphopods, 5 chitons, 8 bivalves, and 10 gastropods.

## Acknowledgements

We thank Carola Greve, Damian Baranski, Alexander Ben Hamadou, and Charlotte Gerheim of the Translational Biodiversity Genomics (TBG) project based in the Senckenberg Research Institute and Museum, Frankfurt, for support with lab work and sequencing. We thank the staff of the Queen's University Marine Laboratory, Portaferry, N Ireland, and Chong Chen (JAMSTEC) for support with field work and specimen collection. This is contribution number 28 of the Senckenberg Ocean Species Alliance. The sampling of *Deshayesiella sirenkoi* was carried out under the Marine Scientific Research permit number MSR U2022-047 from the United States government, since South Chamorro Seamount is within the Commonwealth of the Northern Mariana Islands.This study was supported by the Natural Science Foundation of Shandong Province (ZR2023JQ014) and the Fundamental Research Funds for the Central Universities (202172002 and 202241002). This research was further supported by a generous philanthropic donation to the Senckenberg Gesellschaft für Naturforschung that funds the Senckenberg Ocean Species Alliance, and by Leibniz Association project PHENOME (P123/2021).

## Additional information

### Funding

| Funder | Grant reference number | Author |
|---|---|---|
| Natural Science Foundation of Shandong Province | ZR2023JQ014 | Jin Sun |
| Fundamental Research Funds for the Central Universities | 202172002 | Jin Sun |
| Leibniz Association | Project PHENOMEP 123/2021 | Julia D Sigwart |
| Fundamental Research Funds for the Central Universities | 202241002 | Jin Sun |

The funders had no role in study design, data collection and interpretation, or the decision to submit the work for publication.

### Author contributions

Julia D Sigwart, Conceptualization, Funding acquisition, Visualization, Methodology, Writing – original draft, Writing – review and editing; Yunlong Li, Resources, Data curation, Software, Formal analysis, Investigation, Visualization, Methodology, Writing – original draft, Writing – review and editing; Zeyuan Chen, Validation, Methodology, Writing – review and editing; Katarzyna Vončina, Resources, Data curation; Jin Sun, Conceptualization, Data curation, Formal analysis, Supervision, Funding acquisition, Investigation, Visualization, Project administration, Writing – review and editing

### Author ORCIDs

Julia D Sigwart ⬤ http://orcid.org/0000-0002-3005-6246
Yunlong Li ⬤ https://orcid.org/0000-0002-4978-4788
Zeyuan Chen ⬤ https://orcid.org/0000-0001-9407-1747
Katarzyna Vončina ⬤ https://orcid.org/0000-0002-3210-0407
Jin Sun ⬤ https://orcid.org/0000-0001-8002-6881

Reviewer #2 (Public review): https://doi.org/10.7554/eLife.102542.3.sa1
Author response https://doi.org/10.7554/eLife.102542.3.sa2

## Additional files

### Supplementary files

MDAR checklist

Supplementary file 1. Supplementary tables. Table S1. Summary statistics from genome sequencing of four new chiton genomes, including HiFi and Hi-C. Table S2. Genome assemblies and gene-model prediction for four new chiton genomes. Table S3. Comparison of summary data among available chitons genomes; the four new chiton genomes are indicated in bold text. N.B. analyses also incorporate the other chromosome-level genome for *Liolophura japonica*, which was published prior to the present study. Table S4. Nonsyntenic rate between species (summarised by clade in Table S5) following the method described in the Supporting Information appendix. Table S5. Summary information for translocation rate of 25 species of molluscs, organised by taxonomic class.

### Data availability

The four chiton genomes projects have been deposited with the NCBI BioProject, with *Acanthochitona discrepans* in PRJNA1114954, *A. rubrolineata* in PRJNA1114370, *Callochiton septemvalvis* in PRJNA1114372, and *Deshayesiella sirenkoi* in PRJNA1114373. In each project, the whole-genome sequencing data, Hi-C data, RNA-seq data have been affiliated. The genome assembly and gene-model predictions are deposited at figshare (10.6084/m9.figshare.27894189). The constructed

Molluscan Linkage Groups (MLGs) are available, with the conserved and representative sequences and their predictive corresponding locations (https://github.com/ylify/MLGs; copy archived at *Li, 2025*). The commands used in this study have also been deposited on GitHub (https://github.com/ylify/MLGs), including the genome assembly, repeats and coding regions prediction, phylogenomic inference, and mutual best hits and ancient linkage groups detections.

The following datasets were generated:

| Author(s) | Year | Dataset title | Dataset URL | Database and Identifier |
|---|---|---|---|---|
| Chen Z, Sigwart JD | 2024 | Whole genome project for the chiton Acanthochitona discrepans | https://www.ncbi.nlm.nih.gov/bioproject/PRJNA1114954 | NCBI BioProject, PRJNA1114954 |
| Li Y, Sun J | 2024 | Acanthochitona rubrolineata Genome sequencing | https://www.ncbi.nlm.nih.gov/bioproject/PRJNA1114370 | NCBI BioProject, PRJNA1114370 |
| Li Y, Sun J | 2024 | Callochiton septemvalvis Genome sequencing | https://www.ncbi.nlm.nih.gov/bioproject/PRJNA1114372 | NCBI BioProject, PRJNA1114372 |
| Li Y, Sun J | 2024 | Deshayesiella sirenkoi Genome sequencing | https://www.ncbi.nlm.nih.gov/bioproject/PRJNA1114373 | NCBI BioProject, PRJNA1114373 |
| Li Y | 2024 | Still waters run deep: Large scale genome rearrangements in the evolution of morphologically conservative Polyplacophora | https://doi.org/10.6084/m9.figshare.27894189 | figshare, 10.6084/m9.figshare.27894189 |

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

# Appendix 1

## Extended methods

### Sample collection, sequencing, and assembly

In this work, four chiton species were collected for high-quality genome assembly and annotation (***Supplementary file 1, table S1***), including *Deshayesiella sirenkoi* (Lepidopleurida), *Callochiton septemvalvis* (Callochitonida), *Acanthochiton discrepans,* and *A. rubrolineata* (Chitonida). Pathways for *A. discrepans* differ slightly as the complete sequencing and analysis was undertaken in Germany; pathways for *Callochiton septemvalvis* differ slightly because of problems encountered that we attribute to the highly heterozygous genome.

The foot was the target organ for genomic DNA extraction and the Hi-C library. The pooled strategy for RNA-seq was adopted, which included foot (F), perinotum (P), radula sac (R), shell edge (S), and visceral mass (V). The tissues were frozen at –80°C (by liquid nitrogen for *A. rubrolineta*) and kept at –80°C until needed.

Genomic DNA in the foot was extracted following the guidelines of the SDS method. PacBio high fidelity (HiFi) sequencing was adopted in genome assembly, with the platform and outputs shown in ***Supplementary file 1, table S1***. The Ultra-Low-DNA-Input_Library was adopted in *C. septemvalvis* using SMRTbell Express Template Prep Kit 2.0 because of limited DNA yield for genomic sequencing. Genomic DNA of *C. septemvalvis* proceeded viaa next-generation sequencing in paired-end 150 bp mode, which was employed in a phasing step. Raw HiFi reads were extracted from the.bam files using extracthifi version 1.0.0 for *D. sirenkoi, C. septemvalvis, A. rubrolineata,* and DeepConsensus version 1.2.0 pipeline (***Baid, 2022***) for *A. discrepans*. Adapters in the reads were checked and removed using HiFiAdapterFilt version 2.0. The genome survey was finished using jellyfish v2.2.10 and Genomescope v2.0 (***Ranallo-Benavidez et al., 2020***), implemented 21- and 23-mer, which produced the estimated genome size and heterozygosity level for each sample (***Appendix 1—figure 2***).

The qualified HiFi reads (i.e. adapter-free) were processed for de novo genome assembly using hifiasm v0.16.1-r375 (***Cheng et al., 2021***), which is proposed as one of the most time-efficient and optimal assemblers for HiFi reads. Regarding *C. septemvalvis*, the high heterozygosity level hindered the haploid genome assembly, so we adopted the phasing strategy. Specifically, we assembled the genome firstly using hifiasm with the disabled purge duplication (-l 0) and then obtained two pseudo-haploid genomes using khaper based on the feature of 17-mer (***Zhang et al., 2021***). Among them, the genome with better metrics (i.e. completeness and continuity) was selected as the representative genome of *C. septemvalvis*. For the other three chitons, the contig-level assemblies were obtained using hifiasm v0.16.1 (***Cheng et al., 2021***) with the default parameters. The potential contamination in the genome was detected and removed using blobtools v1.1.1 (***Laetsch and Blaxter, 2017***). Duplicated haplotigs and overlaps were removed for *D. sirenkoi, A. discrepans,* and *A. rubrolineata* using Purge_Dups v1.2.6 (***Guan et al., 2020***).

For *D. sirenkoi, C. septemvalvis, A. rubrolineata,* the Hi-C reads were then aligned to the contig-level genome using the classical pipeline as demonstrated in previous work. In short, the valid reads (i.e. chimera from two primary DNA regions) were detected and extracted using HiC-Pro v3.1.0 (***Servant et al., 2015***), which integrated bowtie2 v2.5.1 (***Langmead and Salzberg, 2012***) and samtools v1.16.1 (***Danecek et al., 2021***) for reads mapping and check. They were further processed by the Juicer v1.6 (***Durand et al., 2016***) for the output of merged_nodups.txt and then the contigs were subjected to 3D de novo assembly (3D-DNA) pipeline v201008 (***Dudchenko et al., 2017***) with the default haploid setting for scaffolding. For *A. discrepans*, the Hi-C reads were aligned to the initial genome assembly using Arima Genomics mapping pipeline (https://github.com/ArimaGenomics/mapping_pipeline; ***Arima Genomics, 2024***). In short, the reads were mapped to the reference using BWA-MEM v0.7.17-r1188 (***Li, 2013***), converted to a sorted.bam file, and filtered to keep uniquely mapping pairs. PCR duplicates were removed using Picard v3.0.0 (***Picard, 2019***). The final alignment file and the assembly were then passed to YaHS v 1.1 a-r3 (***Zhou et al., 2023***) for scaffolding in the default mode. Juicebox Assembly Tools (***Dudchenko et al., 2017***) were used to generate and visualize a Hi-C contact map. We manually curated the scaffolded assembly using an editable Hi-C heatmap to improve the assembly's quality and to correct misassembles with Juicebox v1.11.08 (***Durand et al., 2016***).

## Repeat region and gene models prediction

The species-specific repeat element database in the contig-level haploid genome was de novo reconstructed using RepeatModeler v2.0.3 (*Flynn et al., 2020*), which was further adopted to identify and classify repeat regions in the genome using RepeatMasker v4.1.2-p1 (*Smit et al., 2015*). The identified repeat region in the genome was labeled as the lowercase bases, called a soft-masked genome.

The aligner STAR version 2.7.10 a (*Dobin et al., 2013*) was employed to map RNA-seq data into the soft-masked genome. The.bam files were used in BRAKE2 version 2.1.6 (*Brůna et al., 2021*) which was implemented with Augustus version 3.4.0 and GeneMark version version 3.67_lic for the ab inito gene prediction, producing 'augustus.gff3' and 'genemark.gtf' as ab initio predictions. These.bam files were also used in Trinity v2.13.2 and Stringtie v2.1.1 (*Pertea et al., 2015*) for transcripts assembly with genome-guided mode. Besides, the de novo assembled transcripts were obtained using Trinity, which could improve the completeness of transcripts. Both de novo assembled transcripts and genome-guided transcripts from Trinity were aligned to the soft-masked genome using PASA version 2.5.2 and aligner blat version 35, producing 'pasa_aligned.gff3' as one of RNA-based predictions. Proteins from 28 high-quality genomes in metazoa were aligned to the soft-maked genome using miniport v0.5-r179, producing 'proteins.gff3' as the homology-based prodiction. Evidencemodeler v1.1.1 (*Haas et al., 2008*) was employed to merge or integrate the above three predictions into a comprehensive profile of genes, and the weights of evidence were AUGUSTUS for 2, GeneMark for 1, PASA for 8 or 10, Stringtie for 6, homolog from protein for 5. Genes were removed if they were solely supported by Augustus or GeneMark, due to the low confidence in the ab initio predictions. The EVM consensus predictions were compared and updated with the de novo assembled transcripts using PASA v2.5.2 (*Haas et al., 2008*), which also led to the identification of untranslated regions (UTRs) and alternatively spliced isoforms in genes.

A slightly modified pipeline was employed in *A. discrepans*. Around 79 M RNA-seq reads were aligned to the *A. discrepans* genome using STAR v2.7.3a (*Dobin et al., 2013*). The resulting alignment file served as an important support in all three prediction methods. *Ab initio* gene prediction was performed on the repeat-masked assembly with Braker3 (*Gabriel et al., 2024*) using default parameters. Two previously published chiton genomes, for *Acanthopleura granulata* (*Varney et al., 2021*) and *Hanleya hanleyi* (*Varney et al., 2022*) were selected and used for homology-based prediction. First, the proteins of *A. granulata* and *H. hanleyi* were downloaded from NCBI and aligned against the assembled genome using MMseqs2 (*Hauser et al., 2016*) with the parameter "`-e 100.0 s 8.5 --comp-bias-corr 0 --max-seqs 500 --mask 0 --orf-start-mode 1`". The results of homologous alignments were then combined into gene models with the splice site identified using mapped RNA-seq data with GeMoMa v1.9 (*Keilwagen et al., 2019*) using default parameters. Finally, the gene predictions were further sorted and filtered separately using the GeMoMa module GAF with default parameters. For the transcriptome-based prediction, the transcriptome of *A. discrepans* was assembled by both de novo and genome-guided approaches using Trinity v2.15.0 (*Grabherr et al., 2011*). The results were merged and passed to Program to Assemble Spliced Alignments (PASA) v2.5.2 (*Haas et al., 2008*) for gene predictions. In the end, all the predictions were combined into consensus coding sequence models using EVidenceModeler v1.1.1 (*Haas et al., 2008*), with the weighting of each method as 'ab initio 1; homology-based 2, transcriptome-based 8'. In the end, we filtered out the incomplete genes and the predictions without any orthologs or transcripts supports using gFACs v1.1.2 (*Caballero and Wegrzyn, 2019*).

The functional annotations of predicted protein sequences of chitons were obtained by searching public databases, including KEGG orthology by BlastKOALA (*Kanehisa et al., 2016*), gene ontology by blast2go version 5-basic against NCBI non-redundant protein database (nr). General information on assembly, gene model, and annotation is attached in *Supplementary file 1, table S2*.

## Phylogenomic relationships within Mollusca using highquality genomes

The putative orthologous sequences shared among 62 metazoan genomes or transcriptomes (58 within Mollusca and 4 outside) were checked using OrthoFinder version 2.5.4 (*Emms and Kelly, 2019*), and then.fasta files in the 'Orthogroup_Sequences' directory were filtered under the pipeline modified from KM Kocot' work (*Kocot et al., 2017Song et al., 2023*). At first,.fasta files were removed unless at least 75% of taxa in a single file were sampled. The sequences in them were removed if the length was less than 100 amino acids or if they were redundant ones checked by

uniqHaplo.pl. Mafft version 7.508 (*Katoh and Standley, 2013*) was applied to align sequences with the parameter of '`--localpair --maxiterate 1000`' and BMGE version 1.12 (*Criscuolo and Gribaldo, 2010*) was employed to trim ambiguously aligned columns in alignments. The alignments were removed if the overlap in them was less than 20 amino acids and there were less than 75% of taxa sampled, then, each alignment was used to construct 'approximately maximum likelihood' tree using FastTree version 2.1.11 (*Price et al., 2010*) with the setting of '-slow -gamma,' and then the paralogues in alignments were removed using phylopypruner version 1.2.6 (*Thalen, 2018*) with the setting of '`--min-support 0.9 --mask pdist --trim-lb 3 --trim-divergent 0.75 --min-pdist 0.01 --prune LS`', which generated a supermatrix consisting of 4966 partitions. The phylogenetic relationship among them was investigated using IQ-TREE version 2.1.3 (*Minh et al., 2020*) with the '-MFP' model to compute the best-fit model of each partition and 1000 ultrabootstraps to test the topological support. In this work, Solemyida, the earliest divergent clade in Bivalvia, was removed for testing the position of Scaphopoda.

## Identification of ALGs and conserved synteny in chromosome-level genomes

The method for ancient linkage groups (ALGs) identification was reported in the two published works (*Schultz et al., 2023*; *Simakov et al., 2022*; *Simakov et al., 2020*). It should be noted that all the draft genomes for synteny were re-organized in the following ways: (1) the labels of chromosomes in a genome were re-ordered reversely according to their sizes, that is Chr01 equals to the longest chromosome; (2) these unanchored contigs/scaffolds and the corresponding genes in them were out of consideration; (3) the protein from the longest isoform was selected as the representative protein of gene if two or more isoforms in the gene. The identification of multi-way mutual best hit (MBH) clusters of encoding proteins among organisms or genomes is the core part. At first, the reciprocal blast hits between every two genomes were inferred by diamond blastp v2.1.8.162 (*Buchfink et al., 2021*) with the threshold of '-evalue 0.001 -max_target_seqs 50000'. Then, python was engaged in constructing the linkages of the orthologue among genomes (up to six), with the available scripts in Github (https://github.com/ylify/MLGs; copy archived at *Li, 2025*). Of which, only the strict one-to-one linkage was selected, meaning such an orthologue was presented in all genomes. For example, three proteins were shown as protein a in organism A, protein b in organism B, and protein c in organism C. The reciprocal blast hits revealed that (1) protein a and protein b, (2) protein b and protein c, (3) protein a and protein c, so a linkage of proteins a-b-c was defined.

The linkage groups between every two genomes were checked by Fisher's exact test using R package macrosyntR (*El Hilali and Copley, 2023*) with a significant threshold of p-value below 0.001, but it might define the group as insignificant one falsely if the number of linkages in the group was small. Therefore, the manual check of linkage groups between two genomes is needed using pairwise dot plots in JCVI version 1.2.7 (*Tang et al., 2008*), especially for organisms with genome duplication (e.g. *Acthatina fulica* and *Arion vulgaris*) or the recent insertion event (*Pomacea canaliculata*) or the two organisms with far relationships (e.g. Mollusca and Porifera). Coupling with the chromosomal location of proteins from.gff3 file, any potential linkage groups at the chromosome level would be constructed using 'groups_six_species.py' (a specified script). Accordingly, the ancient molluscan linkage groups (MLGs) were extracted based on the significant linkage groups on every two genomes. For example, similar to the definition of MBH, there were three chromosomal associations among three organisms: (1) chromosome 1 in organism A and chromosome 2 in organism B, (2) chromosome 1 in organism A and chromosome 3 in organism C, (3) chromosome 2 in organism B and chromosome 3 in organism C; then, it should be a linkage group from organism A-B-C as chromosome 1-2-3. Finally, the conserved collinearity among organisms was visualized as the vertical lines (linkages) and horizontal tracks (chromosomes in an organism) using JCVI version 1.2.7 (*Tang et al., 2008*). Colors in vertical lines were used to differentiate the linkage groups.

The ancestral states of nodes were predicted to trace the karyotype evolutionary route in Polyplacophora, from the common ancestor of Mollusca to the genus *Acanthochitona*. In detail, the ancient and conserved linkage groups were inferred from seven chromosome-level mollusk assembles, including one snail (*Gibbula magus*), one bivalve (*Mizuhopecten yessonensis*), and five chitons (*A. rubrolineata*, *A. discrepans*, *C. septemvalvis*, *D. sirenkoi*, *Liolophura japonica*). The obtained gene set consisted of 4729 homologs, which could be detected in all 7 assembles and located in 20 linkage groups, with the name of mollusk linkage groups (MLGs). The AGORA v3.1 (*Muffato et al., 2023*) was employed to predict the gene order within linkage groups in ancestors,

resulting in the assignment of genes in Contiguous Ancestral Regions (CARs). Then, the gene sets with the corresponding pseudo-locations in ancestral chromosomes were generated. We did not predict the sequences in the ancestral stats but integrated the proteins in the linkages (i.e. 7 proteins in a single linkage in this work). A customized Python script was used to check the chromosomal similarity between the MLGs and the extant genomes, with slight modifications from the identification of linkage groups. Specifically, diamond blastp was adopted to find the hits between them (MLGs as the query and proteins from genomes as databases). The protein was defined as a significant match against the mollusk ancestors if it was hit by at least 6 of 7 proteins from a linkage in MLGs (evalue <0.001). Similarly, the significant linkage groups were checked by Fisher's exact test as mentioned before. Finally, the result was visualized as the Oxford plot using Matplotlib. The protein sequences and Python script are available on GitHub (https://github.com/ylify/MLGs; copy archived at *Li et al., 2024*). Similarly, the nodes with more than two chromosome-level genomes within Mollusca were also investigated for the chromosome number of their common ancestor. Moreover, the MLGs will help us to identify the ancestral stat and chromosomal events in Mollusca, including (1) the identification of chromosome duplication, fission, insertion, and fusion and (2) the real sense of whole genome duplication. To compare the syntenic changes between two genomes, the color scheme in MLGs was selected to visualize the classification of MBH. If the two proteins in MBH were both found in MLGs, the MBH would be defined as a significant linkage in Mollusca and then colored in oxford plot otherwise it would be in grey.

Translocation rates in species between chromosomes were evaluated as shown below; the non syntenic rate divided by the divergent time (based on calibrations from the fossil record). The non syntenic rate is the ratio of MBH not in a significant linkage group, compared to the total count of MBH. In this study, 25 genomes from four classes were selected to calculated the translocation rate at the inter chromosome level, including 2 scaphopods, 5 chitons, 8 bivalves, and 10 gastropods. The non syntenic rates of pairwise comparison were shown in *Supplementary file 1, table S4*. The divergent time among these four classes was set as 530 Mya according the first appearance of Bivalvia and Gastropoda (*Benton et al., 2009*; *Bieler et al., 2014*). The translocation rate of species was calculated by the mean value of non syntenic rate from the comparison against species from other classes. For example, the translocation rate in *A. discrepans* was the mean value of its comparison with the rest 20 non-chiton species. The results were shown in *Supplementary file 1, table S5*.

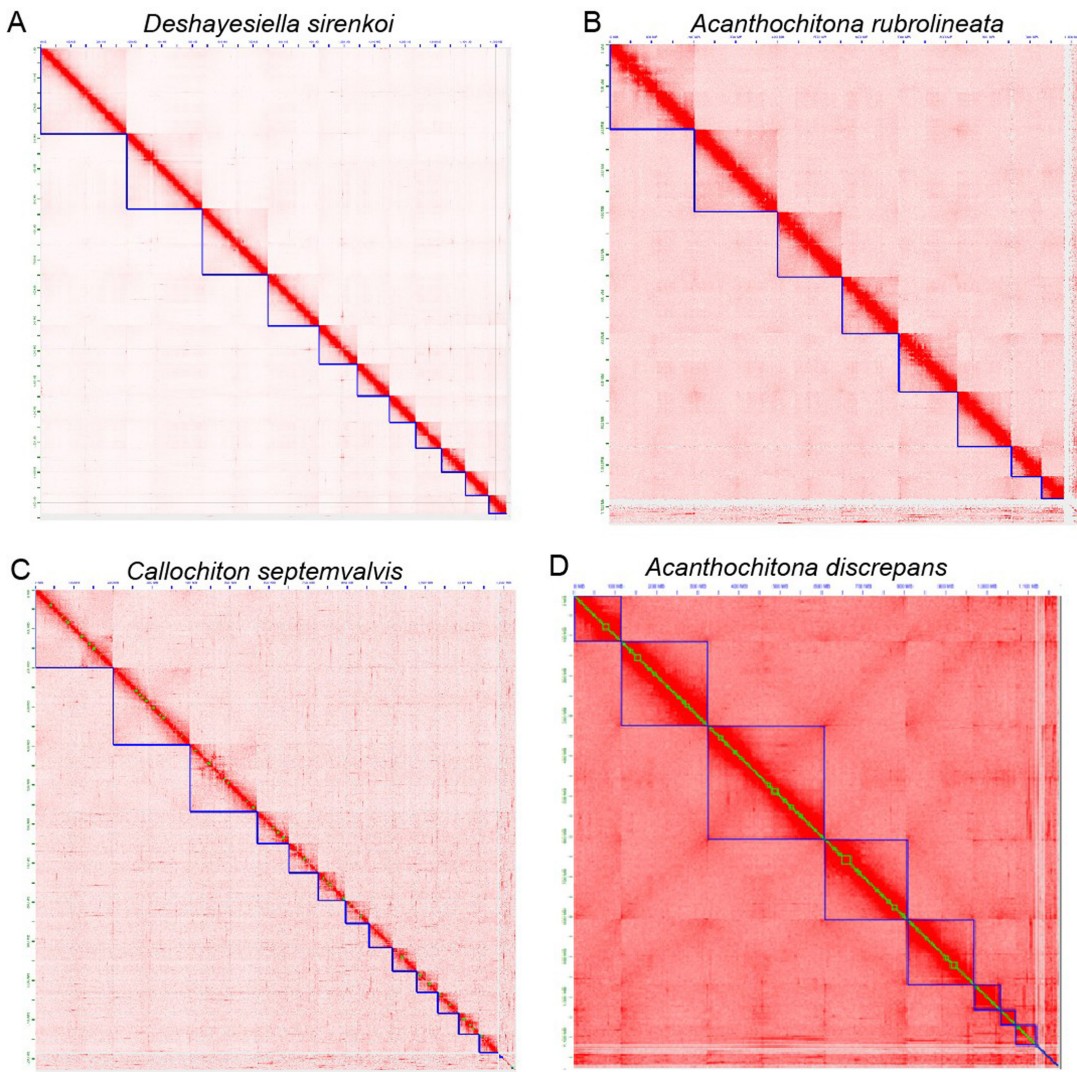

**Appendix 1—figure 1.** Hi-C contact map in pseudo-chromosome level assemblies of four chitons.

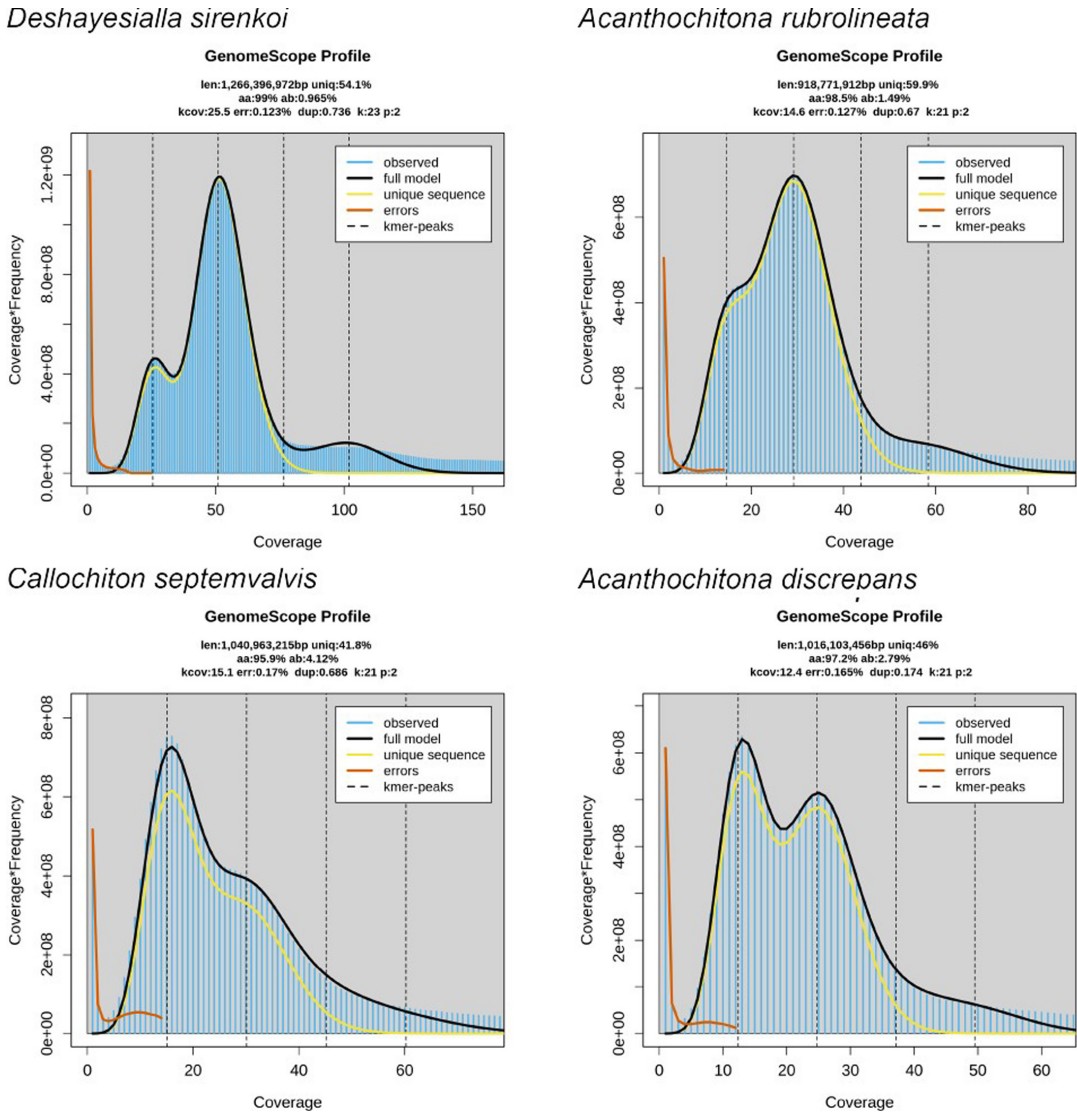

**Appendix 1—figure 2.** Genome survey of four chitons from the GenomeScope2 software.

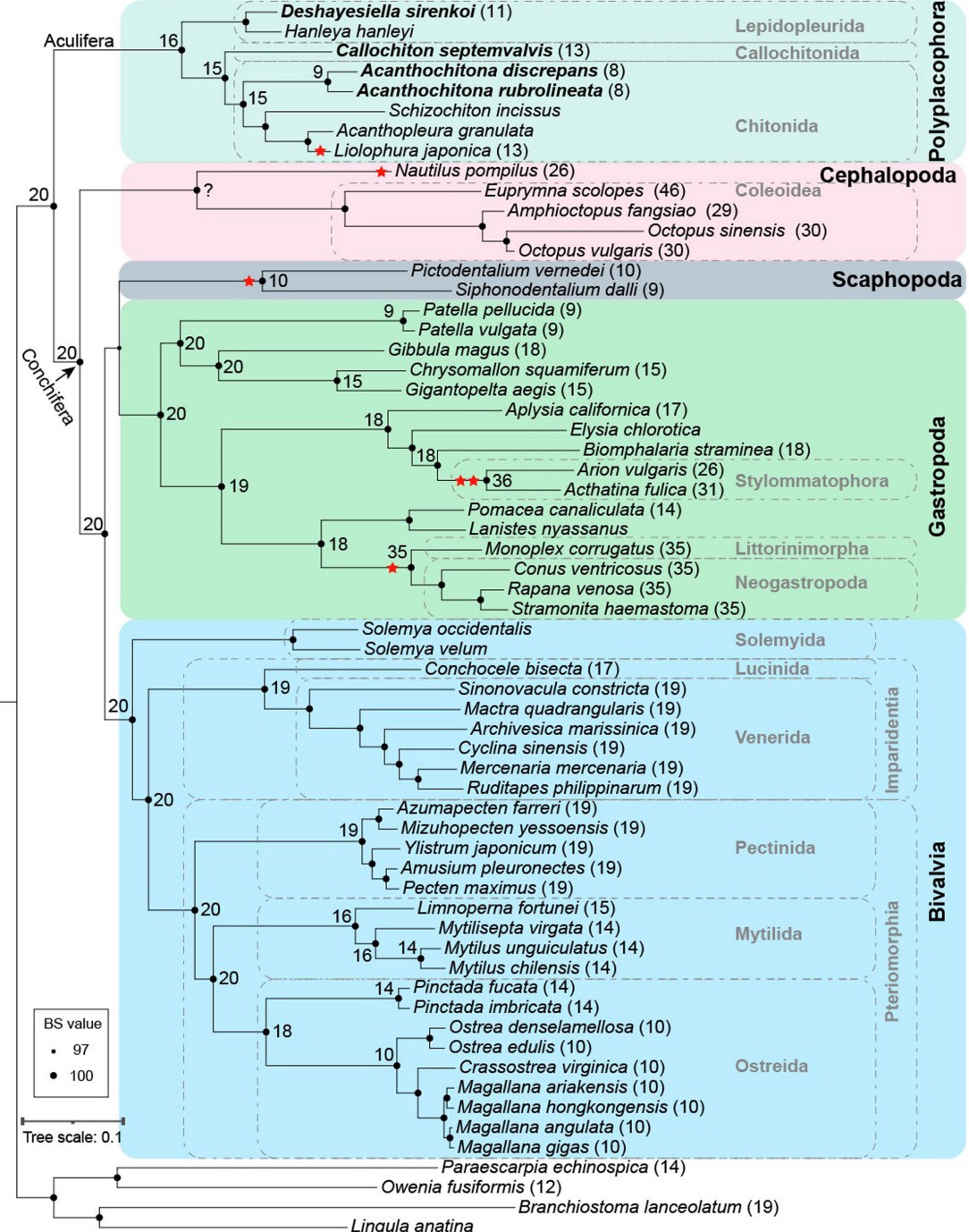

**Appendix 1—figure 3.** Phylogenomic relationships among five classes in Mollusca (with two *Solemya* from RNA-seq). A single star in the tree indicates partial duplications and two stars indicate true whole genome duplication. The numbers in parentheses after taxon names indicate the number of pseudo-chromosomes where known, and the numbers on branches indicate the predicted (1 n) number of ancestral chromosomes. The newly sequenced chiton species are indicated in bold text. In contrast to the tree presented in the main text, this tree resolved Scaphapoda +Gastropoda. IQTREE2: -m MFP -B 1000 (60 genomes and 2 transcriptomes).

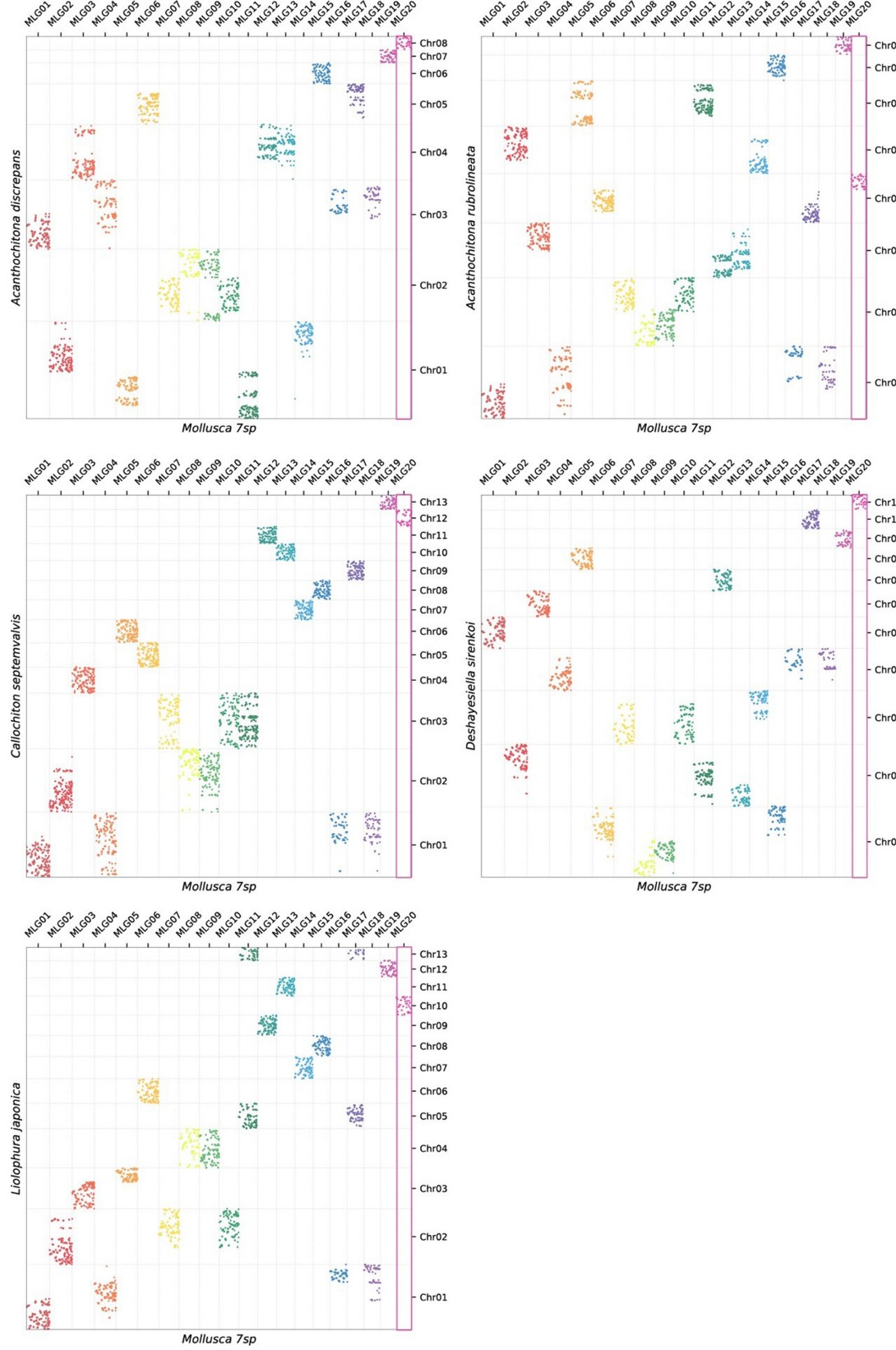

**Appendix 1—figure 4.** Oxford plots of five chitons against MLGs.

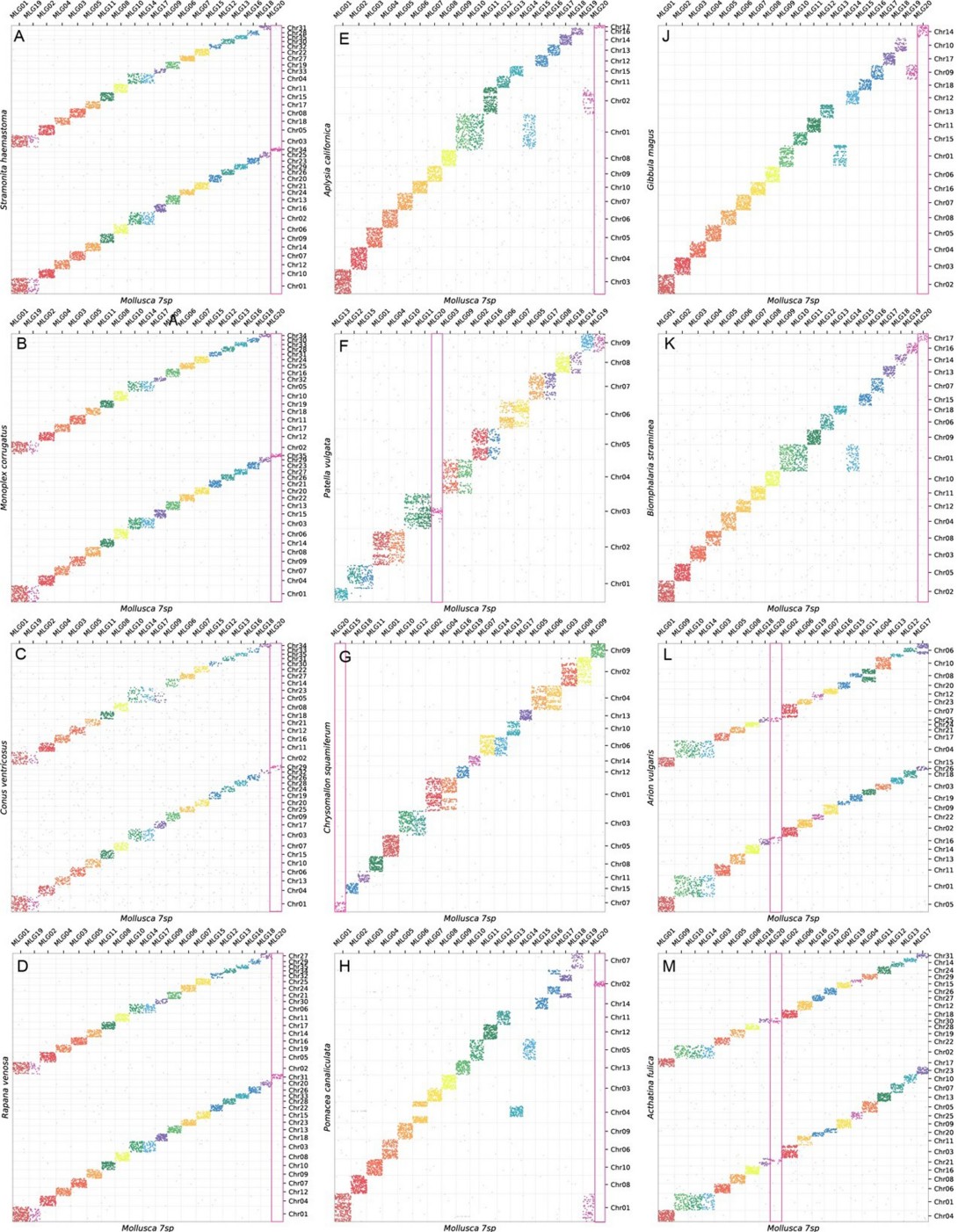

**Appendix 1—figure 5.** Oxford plots of 12 gastropods against MLGs, with the identification of the true sense of whole genome duplications (WGD). MLG20 is highlighted with a red box to distinguish its dynamics under WGD.

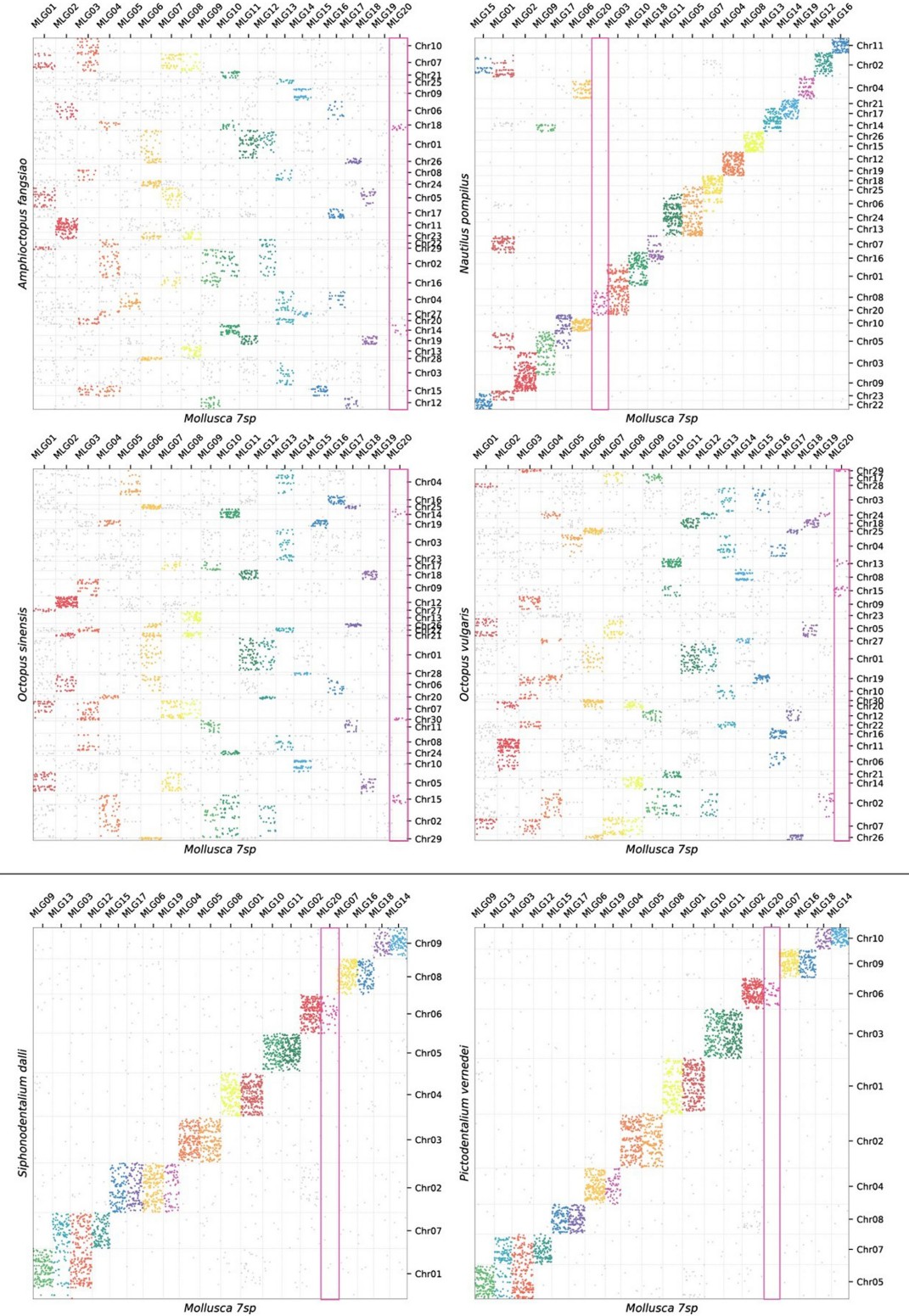

**Appendix 1—figure 6.** Oxford plots of four cephalopods and two scaphopods against MLGs.

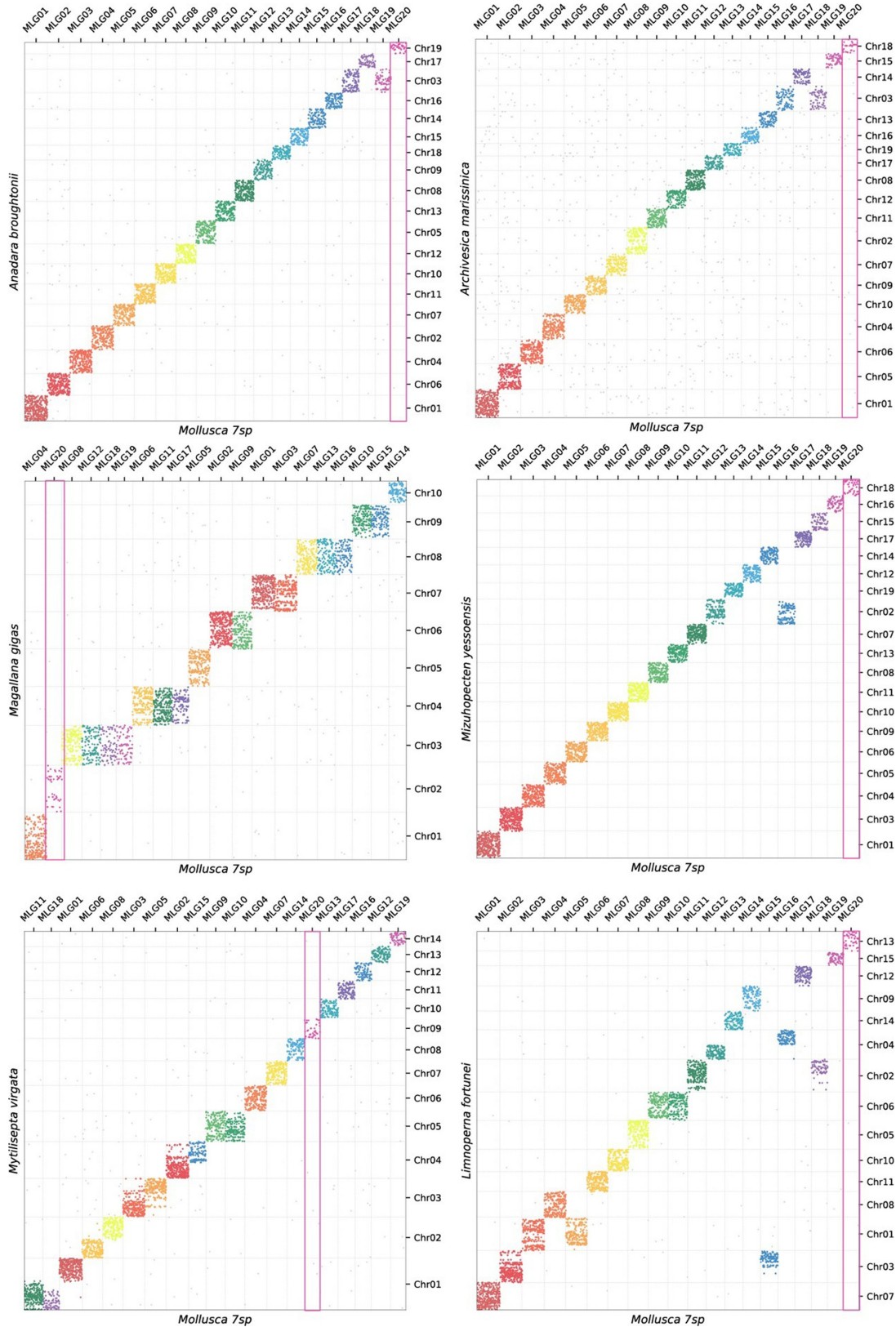

**Appendix 1—figure 7.** Oxford plots of six bivalves against MLGs.

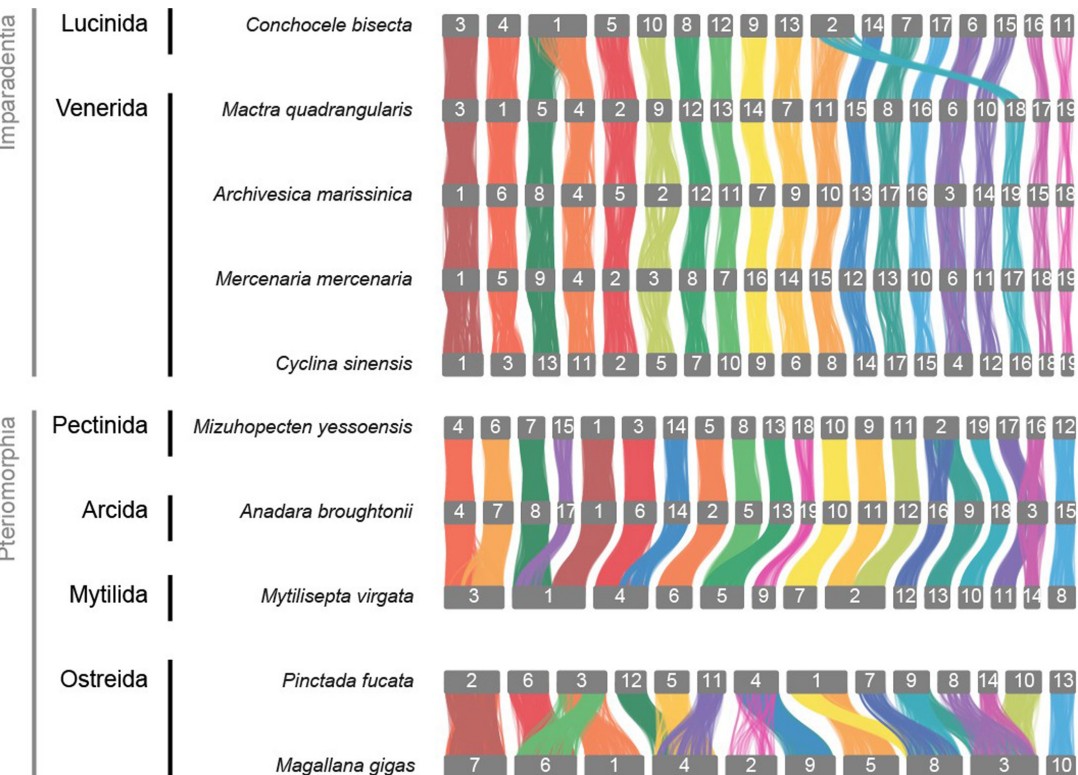

**Appendix 1—figure 8.** Syntenic graph in Bivalvia, highly conserved synteny in Imparadentia and large rearrangements within Peteriomorphia. Two clades in Bivalvia have 19 pseudo-chromosomes from a single fusion, but they are independent occurrences: in Pectinida, the fusion is MLG12 and MLG16, whereas in Venerida, the fusion is MLG16 and MLG18.

