## [Editor Report · eLife Assessment]

This **important** study advances our understanding of genome annotations for chiton genomes. It provides a **solid** estimation of syntentic relationships for the chromosomes of the four new genomes plus an analysis linking these to other available chiton genomes, and an update for how these relate to molluscan genomes.

---

## [Referee Report · Reviewer #2 (Public review)]

In my previous review, I considered the contributions of the authors to be substantial because they have nearly doubled the number of genome sequences for chitons, and their newly sequenced genomes apparently are very well annotated. I would even extend these strengths now that I have had a chance to better review recent literature on marine animal genomes. Their contribution has helped make the chitons one of the best available marine taxa for comparative genomic studies. However, I still am unconvinced by the authors' claims to have demonstrated an unusually high rate of large-scale genome rearrangements across chitons. Their best argument seems to be comparisons drawn within a couple of similarly ancient bivalve lineages that were used to identify the conserved genomic regions in the first place, specifically the 20 molluscan linkage groups (MLGs). Perhaps it is safest to conclude that these MLGs are mostly conserved in conchiferans. Their main comparison with other molluscan classes is presented in tables 4 and 5 in the supplement, where they report a somewhat higher mean translocation rate for chitons (45.48) than for bivalves (41.10) or gastropods (41.87) but does this justify the implications of the title or the claims made in the Summary? I am not sure, partly because these summary tables are not made in a way that separates the gastropod or bivalve species listed into subtaxa separated by LCAs with estimated age, so the mean value across each class is not especially helpful. I still feel that the authors were not convincing in their arguments that chiton chromosomes have been subject to an unexpected history of rearrangement when contrasting quite ancient chitons lineages. This does not include impressive rearrangements documented for the likely geologically recent rearrangements seen within the genus, Acanthochitona, and separately within the subfamily Acanthopleurinae. These are quite impressive events that occurred recently within lineages of shallow-water chiton taxa, not deep still waters.

By the authors' estimates, some sequenced chiton genomes represent lineages that share a last common ancestor (LCA) as much as over 300 million years before present. This is like comparing a frog genome with a human genome. I suspect that the authors would agree that the pace of chiton genome rearrangements is not nearly as great as that observed for younger taxa such as mammals or particular insect orders known to have a history of genome shuffling. For example, according to Damas et al. (2022; https://www.pnas.org/doi/full/10.1073/pnas.2209139119) for comparisons within mammals, "94 inversions, 16 fissions, and 14 fusions that occurred over 53 My differentiated the therian from the descendent eutherian ancestor."

It is more interesting to me how the chiton genome rearrangements compare with other molluscan classes or for comparisons of other marine taxa genomes that share a similarly ancient LCA, but this is difficult to dig out of the authors' presentation. As far as I am aware, there are relatively few such comparisons of genome rearrangements available for marine animals. Attempting to do my own search for any comparison I could make, I noticed in that in a recent compilation of "high quality"* genomes (Martínez-Redondo 2024; https://doi.org/10.1093/gbe/evae235), this included genomes for 84 (mostly insect) arthropods, 67 vertebrates, 31 mollusks, 15 annelids, 12 nematodes, and 6 cnidarians, but the numbers drop off to 1-4 for many phyla, e.g., echinoderms. If there are really so few marine taxa available for comparison to the last 300 My of chiton genome rearrangements and fusions, then I would like to see a better presentation of the contrasts of the 20 molluscan linkage groups (MLGs) across molluscan classes. I found it very difficult to evaluate beyond the assertion that these are relatively conserved in two bivalve taxa. I remain unconvinced whether the amount of genome rearrangement observed by the authors for chitons is either especially rapid or slow. Certainly the genome browsers I have seen do not make it easy to compare, for example, marine gastropod or bivalve genomes (e.g., https://www.ncbi.nlm.nih.gov/cgv/9606 or https://genome.ucsc.edu/cgi-bin/hgGateway).

An unrelated topic that I also brought up in my earlier review is the ancestral reconstruction of molluscan chromosome numbers. The authors' explanation does nothing to justify how they ended up with an optimization of 20 for the ancestor of Mollusca. The outgroups included two annelids, Owenia [12 chromosomes] and Paraescarpia [14], plus the very distant chordate, Branchiostoma [19] (but the tunicate, Oikopleura has 6). Do the authors not understand that outgroups are critical for the optimization of character states at an ancestral node, with the most proximal outgroups being most important? How did they end up with an ancestral reconstruction of the chiton LCA with 16 chromosomes when there is no chiton with more than 13? Given the number of chromosomes in annelids, which is clearly the most proximal outgroups included with chromosome numbers available, it is more parsimonious to postulate that there was an increase in chromosome number for the conchiferan lineage. Related, they should have rooted that tree figure (Fig. 2) with the deuterostome, Branchiostoma, not a monophyletic grouping of all outgroups.

A recent study by Lewin et al. (2024; https://doi.org/10.1093/molbev/msae172) comparing annelid genomic rearrangements suggests to me that annelids probably have a more striking history of rearrangements than for chitons, but I found it difficult to evaluate. I do tend to agree with the overall conclusion of Lewin et al: "All animals with bilateral symmetry inherited a genome structure from their last common ancestor that has been highly conserved in some taxa but seemingly unconstrained in others." That is also my impression so far but the authors have done little to summarize what is known. One study that implies that at least deuterostomes have conserved elements of an ancestral chromosomal arrangement is provided by Lin et al. (2024; https://doi.org/10.1371/journal.pbio.3002661), who sequenced two genomes representing crown group hemichordates (LCA about 504 My).

Even if my general impression is wrong that the history of chiton genome rearrangement is not especially remarkable, or at least we still do not have a great idea of how rapid it is, I still think the authors could have done a better job of demonstrating their claims. This is important if they are going to make big claims about the pace of chiton chromosomal rearrangements. There is very little discussion of other similarly ancient marine animal taxa. I do not especially have a problem with excluding better known terrestrial mammalian or insect genomes as perhaps not a very relevant contrast, but am I supposed to be impressed with the comparisons made with bivalves and gastropods in Tables 4 and 5 of the Supplement? Where do the authors present a detailed comparison of how these estimates compare to conchiferans? Is this amount of genome rearrangement observed for chitons surprising for an extant taxon that has diversified for over 300 My? This is claimed in the title and summary of the manuscript as the take-home for the contribution, but I am left with the impression that there is too little attempt to justify that the pace across Polyplacophora (Neoloricata) is in any way remarkable. Apparently, there are few other lineages of marine taxa within which there are sequenced and well annotated genomes that can be compared, and this confounds the extent of conclusions that can be made.

* "high quality" genomes defined as follows by Martínez-Redondo (2024): "...we lowered the threshold used to consider a data set as high quality to 70% C + F (complete plus fragmented) BUSCO score (Manni et al. 2021), as the original 85% threshold was too restrictive when prioritizing a wide taxonomic sampling and the inclusion of biologically interesting species that are not widely studied."

---

## [Author Response]

The following is the authors’ response to the original reviews.

**Recommendations for the authors:**

**Reviewer #1 (Recommendations for the authors):**
This paper provides a compelling analysis of chiton genomes, revealing extensive genomic rearrangements despite the group's apparent morphological stasis. By examining five reference-quality genomes, the study identifies 20 conserved molluscan linkage groups that are subject to significant rearrangements, fusions, and duplications in chitons, particularly in the basal Lepidopleurida clade. The high heterozygosity observed adds complexity to genome assembly but also highlights notable genetic diversity.

We also note the comment from this reviewer that “more information is needed to clarify how this affects genome assembly and evolutionary outcomes.” We strongly agree; although it is outside the scope of this study, this may help develop future work on that topic.

The research challenges the assumption that morphological stability implies genomic conservatism, suggesting that dynamic genome structures may play a role in species diversification. Although limited by the small number of molluscan genomes available for comparison, this study offers valuable insights into evolutionary processes and calls for further genomic exploration across molluscan clades. Some minor comments need to be tackled:(1) Line 39: 'major changes'. Please, better explain what you mean here?

Clarified as major morphological change

(2) Lines 70-73: refer to 'extant' cephalopods.

Corrected

(3) There is an inconsistency in the use of "Callochitonida" (lines 76, 85, 140, 145, Table S3, Figure S3) and "Chitonida s.l." (Figures 2, 3, and 4) throughout the text, figures, and supplementary material. To maintain clarity and avoid confusion, I recommend choosing one taxon and using it consistently across all sections of the manuscript. This will ensure coherence and help readers follow the discussion without ambiguity.

An explanation has been added to the introduction and other instances in the text changed to Chitonida s.l. for consistency

(4) Overall, the conclusions introduce several important topics and additional information that were not addressed earlier in the paper. It would enhance the coherence and impact of the study to introduce these points in the introduction, as they highlight the broader significance and relevance of the research. Integrating these key aspects earlier on would better frame the study's objectives and provide readers with a clearer understanding of its importance from the outset.

The paragraph about chiton natural history and some additional lines have been moved to the introduction

(5) Lines 242-245 and 254-256: While I agree with the authors on the remarkable results found in molluscs, particularly in polyplacophorans, I suggest toning down the comparisons with lepidopterans. The current framing may come across as dismissive towards butterflies, which does not seem necessary. It's true that biases exist in studying taxa that are more charismatic due to factors like diversity or aesthetic appeal, but the goal should be to emphasize the value of polyplacophorans without downplaying the significance of butterfly research. Instead, the focus should be on highlighting chitons as an exciting new model for understanding key evolutionary processes like synteny, polyploidy, and genome evolution. This shift would underscore the importance of polyplacophorans in a positive light without diminishing the value of lepidopteran studies.

This sentence has been rephrased to adjust the tone of this paragraph

(6) Figure 3: should be read 'Polyplacophora'.

Corrected

**Reviewer #2 (Recommendations for the authors):**
I hope these comments by line number are helpful, despite my lack of experience with comparative genomics:

We note the general comment from this reviewer that “most chiton genomes seem to be relatively conserved” may be a misunderstanding from our presentation; we have added some additional notes in the first part of the discussion to ensure that this is clear to all readers.

The reviewer also pointed out that “geologically recent events that do not especially represent the general pattern of genome evolution across this ancient molluscan taxon”. To clarify, the (limited) phylogenetic evidence suggests these changes are a longer term pattern throughout chiton evolution, since chromosomal rearrangements are found when comparing congeneric species (*Acanthochitona* spp., Fig 4C) and also across orders (Fig 4B). This has been added to the conclusions, as this is clearly an important point that was not adequately explained in the original text.

(1) Line 72: It is true that adaptive radiations occur and are an interesting general model for how diversification can lead to species-rich taxa. However, there are other "non-adaptive" processes that can lead to geographically isolated species that are not much differentiated in their ecological or morphological diversity. The sentence here implies that such adaptive radiation is a necessary correlation of species richness. I agree that chitons have hardly frozen in time since the Paleozoic.

This is clarified by moving some additional natural history aspects of chitons to the introduction, also as suggested by the first reviewer

(2) L113: I am curious about how this character optimization was accomplished to allow the authors to reconstruct the HAM (hypothetical ancestral mollusc) chromosome number as 20 when the range of variation in Polyplacophora is 6 to 16 (mode 11), and chitons are part of the sister taxon to conchiferans. Is this dependent on the chromosome numbers found in the outgroup?

We inferred ancestral linkage groups (“chromosomes”) based on comparison with other gastropods and bivalves noted in the methods; the other study cited (Simakov et al. 2022) used a broader selection of metazoans and also predicted an ancestral Mollusca karyotype of 1N=20.

(3) L116: "Using five chromosome-level genome assemblies for chitons, we reconstructed the ancestral karyotype for Polyplacophora (more strictly the taxonomic order Neoloricata), and all intermediate phylogenetic nodes to demonstrate the stepwise fusion and rearrangement of gene linkage groups during chiton evolution (Fig. 3)."This is probably fine, but I had to struggle to understand what genome events happened between the Acanthochitona species. Are the chromosomes merely ordered and numbered by chromosome size and the switch in position between chromosomes 1 and 3 just has to do with the chromosomes 4+5, so they become the largest chromosome, and the former 1 is now 3? Confusing! The way it is drawn it seems like this implies more genome rearrangement than occurred, whereas if the order was maintained it would be more obvious that there were simply two chromosome fusions.

The linkage groups are numbered in order of size, which is the typical way they would each be presented if the taxon was illustrated alone. Here this allows the reader to understand how the fusions or rearrangements have shifted the volume of genetic information between groups especially in comparison to the molluscan or polyplacophoran ancestor. In Fig 4 we instead decided to present the linkage groups in a revised form, so that each transition from the nearest ancestor is visible in more detail. We have added these points in the figure caption for Fig 3 which should make it easier for new readers to understand the presentation.

(4) L481: Typo: A. rubrolineatain should be A. rubrolineata.

Corrected

(5) Figure 4: I am a little confused with what is meant by an "Ancestor" in these diagrams. For example, for comparing the two species of Acanthochitona with a hypothetical ancestor, it seems that the ancestor should be like one of the two, not different from both.I am looking at Ancestor "3" compared with the Acanthochitona rubrolineata "3" and A. discrepans "4". Again, I assume that the latter is "4" because it is slightly smaller than a new "3" and now the new "3" corresponds to "1" in the other Acanthochitona. This figure does help interpret Figure 3.

To the point about reconstructing ancestral types; the two species both descended from a common ancestor. In morphology it is sometimes clear that one lineage retains more plesiomorphic character states; but in this case we must assume equal probability of change in any direction. The ancestor is a compromise that estimates the shortest distance to both descendants.

We understand how the numbers were unclear and potentially distracting. This has been added to the figure caption, we are grateful for the feedback that will certainly help future readers.